# How Well Do Supervised 3D Models Transfer to Medical Imaging Tasks?

**Wenxuan Li    Alan Yuille    Zongwei Zhou**[*]
Johns Hopkins University
https://github.com/MrGiovanni/SuPreM

## Abstract

The pre-training and fine-tuning paradigm has become prominent in transfer learning. For example, if the model is pre-trained on ImageNet and then fine-tuned to PASCAL, it can significantly outperform that trained on PASCAL from scratch. While ImageNet pre-training has shown enormous success, it is formed in 2D, and the learned features are for classification tasks; when transferring to more diverse tasks, like 3D image segmentation, its performance is inevitably compromised due to the deviation from the original ImageNet context. A significant challenge lies in the lack of large, annotated 3D datasets rivaling the scale of ImageNet for model pre-training. To overcome this challenge, we make two contributions. Firstly, we construct AbdomenAtlas 1.1 that comprises 9,262 three-dimensional computed tomography (CT) volumes with high-quality, per-voxel annotations of 25 anatomical structures and pseudo annotations of seven tumor types. Secondly, we develop a suite of models that are pre-trained on our AbdomenAtlas 1.1 for transfer learning. Our preliminary analyses indicate that the model trained only with 21 CT volumes, 672 masks, and 40 GPU hours has a transfer learning ability similar to the model trained with 5,050 (unlabeled) CT volumes and 1,152 GPU hours. More importantly, the transfer learning ability of supervised models can further scale up with larger annotated datasets, achieving significantly better performance than preexisting pre-trained models, irrespective of their pre-training methodologies or data sources. We hope this study can facilitate collective efforts in constructing larger 3D medical datasets and more releases of supervised pre-trained models.

## 1 Introduction

Pre-training and fine-tuning is a widely adopted transfer learning paradigm (Zoph et al., 2020). Given the relationship across different vision tasks, a model pre-trained on one dataset is expected to benefit another. Over the past few decades, pre-training has been important in AI development (Kumar, 2017; Radford et al., 2021). For 2D vision tasks, there are two available options: (*i*) supervised pre-training and (*ii*) self-supervised pre-training, but for 3D vision tasks, option (*i*) is often not available simply due to the lack of large, annotated 3D volumetric datasets (Wang et al., 2022).

Supervised pre-training can learn image features that are transferable to many target tasks. It has been common practice to pre-train models using ImageNet and then fine-tune the model on target tasks that often have less training data, e.g., PASCAL. However, two challenges arise in ImageNet pre-training. Firstly, ImageNet predominantly comprises 2D images, leaving a palpable void in large-scale 3D datasets and investigation in 3D transfer learning (Huang et al., 2023). Secondly, ImageNet is intended for image classification, so the benefit for segmentation (and other vision tasks) can be somewhat compromised (He et al., 2019). If such an ImageNet-like dataset exists—formed in 3D and annotated per voxel—supervised pre-trained models are expected to transfer better to 3D image segmentation than self-supervised ones for two reasons.

1. **Supervised pre-training is more efficient in data and computation because of its explicit learning objective.** While self-supervised pre-training can learn features without manual annotation, it often requires a large corpus of datasets (Xiao et al., 2022). Extracting meaningful

---
[*]Correspondence to Zongwei Zhou (ZZHOU82@JH.EDU).

features directly from raw, unlabeled data is inherently challenging. Unlabeled data have a high degree of redundancy (Haghighi et al., 2020; 2021) and noise (Mahajan et al., 2018), which can complicate the learning process. Therefore, self-supervised pre-training often calls for greater computational resources and time to match the outcomes achieved by supervised pre-training (Chen et al., 2020a; Tang et al., 2022). We have quantified the improved data and computational efficiency from perspectives of both pre-training (Figure 2a; 99.6% fewer data) and fine-tuning (Figure 2b; 66% less computation). Specifically, the model trained with 21 CT volumes, 672 masks, and 40 GPU hours shows transfer learning ability similar to that trained with 5,050 CT volumes and 1,152 GPU hours, highlighting the remarkable efficiency of supervised pre-training.

2. **Supervised pre-training enables the model to learn image features that are relevant to image segmentation.** Self-supervised pre-training must extract images features from raw, unlabeled data using pretext tasks such as mask image modeling (Chen et al., 2019a; Tao et al., 2020; Zhou et al., 2021b; He et al., 2022), instance discrimination (Xie et al., 2020; Chaitanya et al., 2020; Shekoofeh et al., 2021), etc. Despite their efficacy in pre-training, these pretext tasks share no obvious relation to the target image segmentation. In contrast, supervised pre-training uses semantically meaningful annotations (e.g., organ/tumor segmentation) as supervision, with which the model can mimic the behavior of medical professionals—identifying the edge and boundary of specific anatomical structures. As a result, the pre-training is interpretable, and the learned features are expected to be relevant to image segmentation tasks (Zamir et al., 2018; Ilharco et al., 2022; You et al., 2022). We have demonstrated that the learned features can be *direct inference* for organ segmentation on CT volumes collected from hospitals worldwide (Table 3; evaluated on three novel hospitals). The features learned by supervision can also be *fine-tuned* to perform novel class segmentation (unseen in the pre-training) with higher accuracy and less annotated data than the features learned by self-supervision (Table 4; evaluated on 63 novel classes).

This paper seeks to answer the question *how well the model transfers to 3D medical imaging tasks* IF it is pre-trained on large, annotated 3D datasets. Naturally, we start with creating an *IF* dataset at a massive scale. **Firstly**, we construct a dataset (termed AbdomenAtlas 1.1[1]) of 9,262 CT volumes with per-voxel annotations of 25 anatomical structures and pseudo annotations of seven types of tumors. This large-scale, fully-annotated dataset enables us to train models in a fully supervised manner using multi-organ segmentation as the pretext task. As reviewed in Table 1, this dataset is much more extensive (considering both the number of CT volumes and annotated classes) than public datasets (Wasserthal et al., 2022; Ma et al., 2022; Qu et al., 2023). Scaling experiments in §3.1 suggested that pre-training models on more annotated datasets can further improve the transfer learning ability. **Secondly**, we develop a suite of **Su**pervised **Pre**-trained **M**odels, termed SuPreM, that combined the good of large-scale datasets and per-voxel annotations, demonstrating the efficacy across a range of target segmentation tasks. As reported in §3.2, some of the dominant segmentation backbones have been pre-trained and will be available to the public. Current pre-trained backbones are U-Net (CNN-type) (Ronneberger et al., 2015), SegResNet (CNN-type) (Myronenko, 2019), and Swin UNETR (Transformer-type) (Tang et al., 2022), and more backbones will be added along time.

In prospective endeavors, we anticipate that the expansion of datasets and annotations will not only enhance feature learning, as demonstrated in this study, but also promote the development of advanced AI algorithms and benchmark the state of the art in terms of segmentation performance, inference efficiency, and domain generalizability.

## 2 BRIEF HISTORY: SUPERVISED PRE-TRAINING

In a major initiative aimed at developing widely transferable AI models—known as Foundation Models in the medical domain (Moor et al., 2023; Butoi et al., 2023; Ma & Wang, 2023a)—one faces a critical decision: *should the focus of pre-training be supervised or self-supervised?* While human annotations undeniably improve task-specific performance, such as semantic segmentation, the best strategy for learning generic image features that can be transferable across a spectrum of tasks has yet to be determined. For 2D vision tasks, the advent of ImageNet (Deng et al., 2009) makes it possible to debate the merits and limitations of supervised pre-trained models for transfer learning compared

---

[1]Segmentation is fundamental in the medical domain (Ma & Wang, 2023b). It can be viewed as a per-voxel classification task. Therefore, the per-voxel supervision used in our pre-training (**272.7B** annotated voxels) is much stronger than the per-image supervision used in ImageNet pre-training (**14M** images).

with self-supervised ones. We refer the readers to Yang et al. (2020) and Tendle & Hasan (2021) for a plethora of viewpoints from either side. In essence, the debates are about clarifying the learning objective (loss function) of emulating human vision (Zhou, 2021).

The learning objective of supervised pre-training is to minimize the discrepancy between AI predictions and semantic labels annotated by humans. Over the years, supervised pre-training on ImageNet has shown marked success in transfer learning (Yosinski et al., 2014). Moreover, the transfer learning ability can be further enhanced when models are trained on increasingly expansive datasets, such as ImageNet-21K (Kolesnikov et al., 2020), Instagram (Mahajan et al., 2018), JFT-300M (Sun et al., 2017), and JFT-3B (Zhai et al., 2022). In general, supervised pre-training exhibits clear advantages over self-supervised pre-training when sizable annotated datasets are available (Steiner et al., 2021; Ridnik et al., 2021). However, acquiring millions of manual annotations is labor-intensive, time-consuming, and challenging to scale—but certainly not impossible—evidenced by several recent influential endeavors (Kuznetsova et al., 2020; Mei et al., 2022; Kirillov et al., 2023; Bai et al., 2023).

On the other hand, self-supervised pre-training offers an alternative by enabling AI models to learn from raw, unlabeled data (Jing & Tian, 2020; Zoph et al., 2020; Ren et al., 2022; 2023), thus reducing the need for manual annotation. Self-supervised pre-training has historically lagged behind the state-of-the-art supervised pre-training in ImageNet benchmarks (Pathak et al., 2016; Noroozi & Favaro, 2016). The recent pace of progress in self-supervised pre-training has yielded models whose performance not only matches but, at times, surpasses those achieved by supervised pre-training (Chen et al., 2020a; Grill et al., 2020; Chen et al., 2020b; Zhou et al., 2021a; Wei et al., 2022). This has raised hopes that self-supervised pre-training could indeed replace the ubiquitous supervised pre-training in advanced computer vision going forward. The caveat, however, is the significant demand for both data and computational power, often exceeding the resources available in academic settings. For example, He et al. (2020) have demonstrated that self-supervised features trained on 1B images (a factor of $714\times$ larger) can transfer comparably or better than ImageNet features.

Supervised pre-training on ImageNet has demonstrated benefit for 2D medical image tasks after transfer learning (Tajbakhsh et al., 2016; Shin et al., 2016; Zhou et al., 2017). Unfortunately, it has been constrained for 3D medical imaging tasks due to the lack of a 3D counterpart to ImageNet. Although there are a great number of raw, unlabeled medical images available (Team, 2011; Baxter et al., 2023; Zhao et al., 2023; Saenz et al., 2024), annotating these images is a labor-intensive undertaking for professionals. Our contribution to a large, annotated 3D dataset could spark the debate of whether self-supervised or supervised pre-training leads to better performance and data/computational efficiency, which would not be possible without the invention of a dataset of such a scale.

## 3 MATERIAL & METHOD

We constructed an AbdomenAtlas 1.1 dataset comprising **9,262** three-dimensional CT volumes and over **251,323** masks spanning **25** anatomical structures and **7** types of tumors. In addition, we released a suite of supervised pre-trained models (SuPreM) to benefit 3D medical imaging tasks.

### 3.1 EXTENSIVE DATASET: ABDOMENATLAS 1.1

Interactive segmentation, an integration of AI algorithms and human expertise, was used to create AbdomenAtlas 1.1 in a semi-automatic procedure. We recruited a team of ten radiologists to perform manual annotations to ensure the annotation quality[2]. Given the complexity of 3D data, rather than annotating the entire dataset voxel by voxel, we asked the radiologists to focus on the most important CT volumes and regions therein. In doing so, an importance score for each volume was computed, derived from the uncertainty, consistency, and overlap (Qu et al., 2023). Six junior radiologists revised the annotations predicted by AI under the supervision of four senior radiologists, and in turn, AI improved its predictions by learning from these revised annotations. This interactive procedure continued to enhance the quality of annotations until no major revision was required from the radiologists. Subsequently, four senior radiologists went through the final visualizations for all the annotations, detecting and revising major errors as needed before the dataset was released. Annotation tools employed included a licensed version from Pair and an open-source MONAI Label.

---

[2]Ensuring high-quality annotations is costly and time-consuming, yet it is critical for transfer learning, as quantified in Appendix A.3, and for reducing ambiguity when training AI models for image segmentation.

Table 1: **Contribution #1: An extensive dataset of 9,262 CT volumes with per-voxel annotations of 25 anatomical structures.** This dataset is unprecedented in terms of data and annotation scales, providing over 251,323 organ/tumor masks and 2,789,975 annotated images that are taken from 88 hospitals worldwide. In 2009, before the advent of ImageNet (Deng et al., 2009), it was challenging to empower an AI model with generalized image representation using a small or even medium size of labeled data, the same situation, we believe, that presents in 3D medical image analysis today. As seen in the table, the annotations of public datasets are limited, partial, and incomplete, and the CT volumes in these datasets are often biased toward specific populations, medical centers, and countries. Our constructed dataset mitigates these gaps, representing a significant leap forward in the field. The CT volumes in datasets 1–17 are used to construct AbdomenAtlas 1.1. The domain gap across these datasets is illustrated in Appendix A.1.

| dataset (year) [source] | # of organ | # of$^\dagger$ volume | # of center | dataset (year) [source] | # of organ | # of$^\dagger$ volume | # of center |
|---|---|---|---|---|---|---|---|
| 1. Pancreas-CT (2015) [link] | 1 | 42 | 1 | 2. CHAOS (2018) [link] | 4 | 20 | 1 |
| 3. CT-ORG (2020) [link] | 5 | 140 | 8 | 4. BTCV (2015) [link] | 12 | 47 | 1 |
| 5. AMOS22 (2022) [link] | 15 | 200 | 2 | 6. WORD (2021) [link] | 16 | 120 | 1 |
| 7-12. MSD CT Tasks (2021) [link] | 9 | 945 | 1 | 13. LiTS (2019) [link] | 1 | 131 | 7 |
| 14. AbdomenCT-1K (2021) [link] | 4 | 1,050 | 12 | 15. KiTS (2020) [link] | 1 | 489 | 1 |
| 16. FLARE'23 (2022) [link] | 13 | 4,100 | 30 | 17. Trauma Det. (2023) [link] | 0 | 4,711 | 23 |
| 18. AbdomenAtlas 1.0 (2023) [link] | 9 | 5,195 | 26 | 19. AbdomenAtlas 1.1 | 25 | 9,262$^\ddagger$ | 88 |

$^\dagger$Our reported number of CT volumes may differ from original publications, as some CT volumes are reserved for validation purposes.

$^\ddagger$The number of CT volumes in AbdomenAtlas 1.1 is lower than the sum of datasets 1–17 due to overlaps within these public datasets.

AbdomenAtlas 1.1 is a composite dataset that unifies CT volumes from public datasets 1–17 as summarized in Table 1. AbdomenAtlas 1.1 presents a level of diversity because the CT volumes are sourced from 88 hospitals worldwide, including pre, portal, arterial, and delayed phases. The gap between these CT volumes includes changes in image quality due to different acquisition parameters, reconstruction kernels, and contrast enhancement, shown in Appendix A.1. Moreover, we provide per-voxel annotations for 25 anatomical structures, including 16 abdominal organs, two thorax organs, five vascular structures, and two skeletal structures. We also provide pseudo annotations for seven types of tumors, namely liver, kidneys, pancreatic, hepatic vessel, lung, colon tumors, and kidney cysts. In total, more than 272.7B voxels are annotated in AbdomenAtlas 1.1, marking a significant leap compared with the 4.3B voxels annotated in the public datasets, amplifying the annotations by a factor of 63.4× (shown in Appendix Figure 4). The high annotation quality is due to the uniform annotation standards described in Appendix A.2. *We commit to releasing AbdomenAtlas 1.1 to the public.* However, this dataset, the largest public per-voxel annotated CT collection by far, accounts for around 0.01% of the CT volumes annually acquired in the United States (Papanicolas et al., 2018). Therefore, cross-institutional collaboration is crucial for accelerating data sharing, annotation, and AI development (Saenz et al., 2024).

## 3.2 A SUITE OF PRE-TRAINED MODELS: SUPREM

The magnitude of our AbdomenAtlas 1.1 is unprecedented in terms of data and annotations. One of the advantages is that it enables us to train AI models in both a supervised and self-supervised manner. At the time this paper is written, neither supervised nor self-supervised pre-training has been performed on this scale of dataset (9,262 volumetric data)[3]. We have developed models (termed SuPreM) pre-trained on data and annotations in AbdomenAtlas 1.1, which leverage established CNN backbones, such as U-Net and SegResNet, as well as Transformer backbones, such as Swin UNETR. With the growing trend of using pre-trained models, we have maintained a standardized, accessible model repository for sharing public model weights as well as a suite of supervised pre-trained models (SuPreM) released by us. Releasing pre-trained models should be considered a marked contribution as they offer an alternative way of knowledge sharing while protecting patient privacy (Sellergren et al., 2022; Zhang & Metaxas, 2023; Ma et al., 2023a). In this study, all of the models in SuPreM follow pre-training and fine-tuning configurations as below.

---

[3]For supervised pre-training, the largest study to date was by Liu et al. (2023), which was developed on 3,410 (2,100 for training and 1,310 for validation) annotated CT volumes. For self-supervised pre-training, the largest one was by Tang et al. (2022), which was trained on 5,050 unannotated CT volumes. Concurrently, Valanarasu et al. (2023) pre-trained a model on 50K volumes of CT and MRI using self-supervised learning.

Table 2: **Contribution #2: A suite of pre-trained models (termed SuPreM) comprising several widely recognized AI models.** We provide pre-trained AI models based on CNN, Transformer, and their hybrid versions, and more AI models will be added. Each model was supervised pre-trained on large datasets and per-voxel annotations from AbdomenAtlas 1.1. Compared with learning from scratch and publicly available models, fine-tuning the models in SuPreM consistently achieves state-of-the-art organ and tumor segmentation performance on two datasets. All of the results, including the mean and standard deviation (mean±s.d.) across ten trials. In addition, we have further performed an independent two-sample $t$-test between learning from scratch and fine-tuning models in our SuPreM. The performance gain is statistically significant at the $P = 0.05$ level, with highlighting in a light red box. Detailed per-class performance can be found in Appendix §B.1.

| model (# of param) | pre-training | TotalSegmentator v1 | | | proprietary dataset | | |
|---|---|---|---|---|---|---|---|
| | | organ | muscle | cardiac | organ | gastro | cardiac |
| U-Net (2015) family (19.08M) | scratch | 88.9±0.6 | 92.9±0.4 | 88.8±0.7 | 85.6±0.5 | 69.8±1.2 | 38.1±1.1 |
| | Zhou et al. (2019b) | 87.8 | 90.1 | 86.3 | 80.1 | 65.5 | 36.9 |
| | Chen et al. (2019b) | 86.9 | 91.4 | 87.4 | 79.0 | 66.2 | 36.7 |
| | Xie et al. (2022) | 88.5 | 92.9 | 89.0 | - | - | - |
| | Zhang et al. (2021) | 89.3 | 93.8 | 89.1 | 85.7 | 72.7 | 38.3 |
| | **SuPreM** | 92.1±0.3 | 95.4±0.1 | 92.2±0.3 | 90.8±0.2 | 76.2±0.8 | 70.5±0.5 |
| Swin UNETR (2021) (62.19M) | scratch | 86.4±0.5 | 88.8±0.5 | 84.5±0.6 | 77.3±0.9 | 65.9± 1.7 | 35.5±1.4 |
| | Tang et al. (2022) | 89.3 | 93.8 | 88.3 | 87.9 | 72.5 | 38.9 |
| | Liu et al. (2023) | 89.7 | 94.1 | 89.4 | 89.1 | 74.6 | 67.6 |
| | **SuPreM** | 91.3±0.3 | 94.6±0.2 | 90.3±0.3 | 90.4±0.7 | 75.9±1.2 | 69.8±0.9 |
| SegResNet (2019) (4.7M) | scratch | 88.6±0.5 | 91.3±0.4 | 89.8±0.4 | 80.6±0.8 | 67.0±1.4 | 36.0±1.3 |
| | **SuPreM** | 91.3±0.5 | 94.0±0.1 | 91.3±0.5 | 86.6±0.3 | 73.7±1.0 | 67.9±0.8 |

To perform a fair and rigorous comparison, we benchmarked with public pre-training methods by pre-training SuPreM using 2,100 CT volumes (same as Liu et al. (2023) and fewer than Tang et al. (2022)) in Tables 2, 4 and Figures 1, 2b, 3. Then, we scaled up the number of CT volumes for pre-training to 9,262 CT volumes to perform direct inference in Table 3. Lastly, we scaled down the number of CT volumes to 21 to explore the edge of our SuPreM in Figure 2a. All these pre-trained models and configurations have been summarized in Appendix Table 8. The best-performing model was selected based on the highest average DSC score over 32 classes on a validation set of 1,310 CT volumes. Implementation details of both pre-training and fine-tuning can be found in Appendix B.2.

The transfer learning ability is assessed by segmentation performance on two datasets, i.e., TotalSegmentator v1 and a proprietary dataset. Benchmarking results in Table 2 indicate that, in comparison with learning from scratch and with existing public models, those fine-tuned from our SuPreM consistently attain superior organ, muscle, cardiac, and gastro segmentation performance on both datasets. U-Net, as a simple and lightweight segmentation backbone, still performs competitively compared with alternative choices like Swin UNETR. This observation is aligned with the majority of the medical imaging community (Isensee et al., 2021; Eisenmann et al., 2023), suggesting that more exploration is needed for advancing segmentation backbones. Moreover, in the scenarios of either small data regimes shown in Figure 1 or large data regimes shown in Appendix Figure 7a–d, supervised models transfer better than their self-supervised counterparts. In summary, our SuPreM surpasses all existing 3D pre-trained models by a large margin in transfer learning performance, irrespective of their pre-training methodologies or data sources.

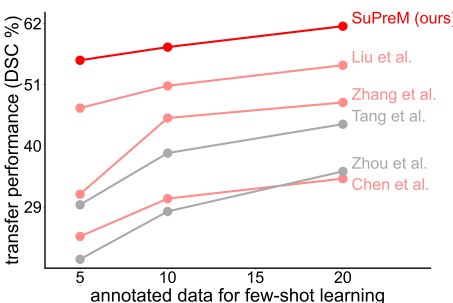

Figure 1: We present the transfer performance on a proprietary dataset with few-shot examples ($N = 5, 10, 20$). The transfer performance (Y-axis) stands for the average DSC score across 20-class organ segmentation and 3-class tumor segmentation. Generally speaking, in a few-shot learning setting, supervised pre-trained models (in red) transfer better than self-supervised pre-trained models (in gray). Notably, our SuPreM achieves the best transfer performance over other well-known publicly available models.

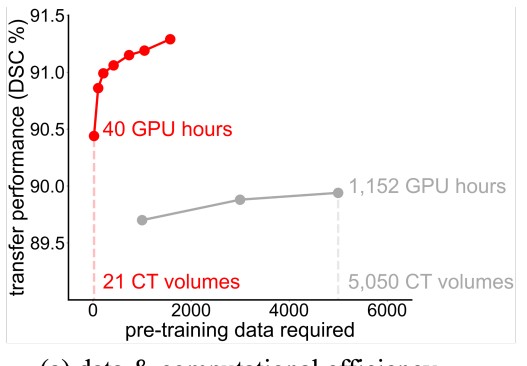 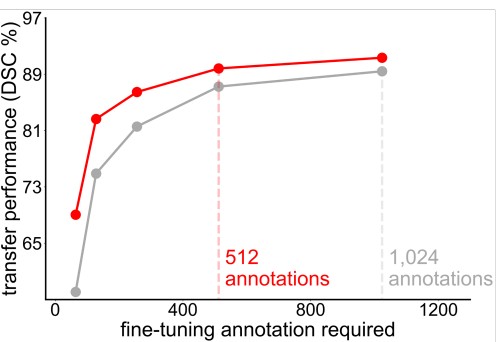

(a) data & computational efficiency
in *pre-training*

(b) annotation & learning efficiency
in *fine-tuning*

Figure 2: **Analysis of pre-training and fine-tuning efficiency.** For a fair comparison, both supervised (in red) and self-supervised (in gray) models use Swin UNETR as the backbone, and the compared self-supervised pre-training is the current state of the art (Tang et al., 2022). The target task was on TotalSegmentator v1. **(a)** scales the model transfer learning ability when pre-trained on varying numbers of images. The results indicate a consistent improvement in transfer learning ability when pre-training on more images. The model trained with 21 CT volumes, 672 masks, and 40 GPU hours shows a transfer learning ability similar to that trained with 5,050 CT volumes and 1,152 GPU hours. Specifically, supervised pre-training is more efficient, requiring 99.6% fewer data and 96.5% less computation. **(b)** assesses the annotation & learning efficiency by fine-tuning models on different numbers of annotated CT volumes from TotalSegmentator. Specifically, SuPreM, fine-tuned on 512 per-voxel annotated CT volumes, can achieve a segmentation performance on par with self-supervised models fine-tuned on 1,024 volumes, reducing 50% manual annotation cost for target tasks.

## 4 EXPERIMENT & ANALYSIS

### 4.1 DATA, ANNOTATION, AND COMPUTATIONAL EFFICIENCY

***Summary.*** We demonstrate the remarkable efficiency: (1) SuPreM trained with 21 CT volumes, 672 masks, and 40 GPU hours shows transfer learning ability similar to that trained with 5,050 CT volumes and 1,152 GPU hours. (2) SuPreM requires 50% fewer manual annotations for organ/tumor segmentation than self-supervised pre-training.

***Data efficiency*** for pre-training. As shown in Figure 2a, supervised pre-training requires less data (21 vs. 5,050 CT volumes) for the pretext task than self-supervised pre-training. This discrepancy arises from the inherent differences in their learning learning objectives and the information they leverage. Supervised pre-training benefits from explicit annotations, which provide direct guidance for the task, i.e., segmentation in this study. The model learns features from both data and annotations, which offer strong and precise supervision. On the other hand, self-supervised learning relies on pretext tasks derived from the raw data, which may offer a more ambiguous learning signal, therefore requiring more examples to capture meaningful features. Importantly, our finding suggests that supervised pre-training is more scalable with increased data. When data are increased from 21 to 1,575 volumes, the transfer learning performance on TotalSegmentator improves from 90.4% to 91.3%. In comparison, for self-supervised pre-training, an increase in data from 1,000 to 5,050 volumes only marginally improves performance from 89.7% to 89.9%. Therefore, supervised pre-training requires significantly less data than self-supervised and is more scalable and effective with increased data.

***Annotation efficiency*** for fine-tuning. We have assessed the annotation efficiency by fine-tuning SuPreM and self-supervised models (Tang et al., 2022) on the TotalSegmentator dataset. Figure 2b suggests that fine-tuning SuPreM can reduce annotation costs for the segmentation task by 50%, averaged over the classes that were not used for pre-training (per-class performance can be found in Appendix Figure 8a–d). Specifically, SuPreM fine-tuned on 512 per-voxel annotated CT volumes can achieve segmentation performance similar to Tang et al. (2022) fine-tuned on 1,024 annotated CT volumes. The fine-tuning performance improvement gets bigger when the number of annotated

Table 3: **Direct inference on three external datasets.** We conduct external validation across four hospitals worldwide. Specifically, our SuPreM—trained on 9,262 CT volumes—is directly inferred on three external datasets, i.e., TotalSegmentator (representing the Central European population from Switzerland; one hospital), DAP Atlas (the Central European population from Germany; two hospitals), and the proprietary dataset (the North American population from the United States; one hospital) measured by DSC scores. For every dataset, we compare the *out-of-distribution* (OOD) performance obtained by SuPreM with *independently and identically distributed* (IID) performance obtained by AI models directly trained on that specific dataset, which are often considered as upper bound performance in domain transfer literature. We find that SuPreM can be generalized well across external datasets without additional fine-tuning, yielding comparable or even superior performance to the IID counterparts, evidenced by the one-sample $t$-test results. Appendix D.1 provides visual examples of anatomical structure segmentation.

| class | TotalSegmentator v1 | | DAP Atlas | | our proprietary dataset | |
|---|---|---|---|---|---|---|
| | SuPreM (OOD) | Liu et al. (IID) | SuPreM (OOD) | Jaus et al. (IID) | SuPreM (OOD)) | Wang et al. (IID) |
| spleen | $96.0\pm0.0$ **** | 93.6 | $96.8\pm0.0$ ns | 96.8 | $95.0\pm0.0$ **** | 89.6 |
| kidney right | $93.3\pm0.1$ * | 94.1 | $96.3\pm0.1$ **** | 95.3 | $92.2\pm0.0$ **** | 88.0 |
| kidney left | $91.2\pm0.2$ **** | 87.7 | $96.4\pm0.1$ **** | 97.4 | $91.6\pm0.1$ **** | 83.9 |
| gall bladder | $81.8\pm0.3$ **** | 73.9 | $87.6\pm0.4$ **** | 71.2 | $83.6\pm0.2$ ns | 85.4 |
| liver | $96.4\pm0.1$ ns | 96.8 | $97.3\pm0.1$ **** | 98.5 | $95.0\pm0.3$ **** | 91.4 |
| stomach | $87.3\pm0.3$ ns | 89.2 | $95.3\pm0.2$ **** | 96.1 | $92.2\pm0.1$ * | 93.6 |
| aorta | $80.8\pm0.4$ **** | 90.7 | $90.7\pm0.5$ **** | 97.7 | $73.9\pm0.3$ **** | 87.0 |
| postcava | $77.9\pm0.3$ **** | 82.1 | $89.1\pm0.4$ **** | 95.9 | $77.7\pm0.4$ ** | 80.8 |
| pancreas | $84.6\pm0.2$ **** | 80.8 | $90.6\pm0.2$ **** | 93.7 | $79.0\pm0.3$ ns | 79.3 |
| **average** | $87.7\pm0.2$ ns | 87.6 | $93.3\pm0.2$ **** | 93.6 | $86.7\pm0.2$ ns | 86.1 |

ns $P > 0.05$    * $P \leq 0.05$    ** $P \leq 0.01$    *** $P \leq 0.001$    **** $P \leq 0.0001$

CT volumes is limited in the target task (e.g., 64, 128, 256). In addition, similar levels of annotation efficiency (reduced 50% cost) are observed when fine-tuning SuPreM on the three-class tumor segmentation task using the proprietary dataset, as presented in Appendix Figure 8e–g.

***Computational efficiency*** for both pre-training and fine-tuning. This efficiency stems, in part, from the reduced data requirements inherent to supervised pre-training, as discussed above. As shown in Figure 2a, supervised pre-training only needs 40 GPU hours to achieve a transfer learning performance comparable to that of self-supervised pre-training, which requires 1,152 GPU hours—a factor increase of 28.8×. When fine-tuning on target tasks, such as on a 10% subset of TotalSegmentator in Appendix Figure 9, the supervised pre-trained model converges much faster than the self-supervised one, reducing the GPU hours needed from 60 to 20. This implies that image features learned by supervised pre-training are intrinsically more expressive, enabling the model to seamlessly adapt across a myriad of 3D image segmentation tasks with minimal annotated data for fine-tuning. This computational efficiency makes supervised pre-training a compelling choice for 3D image segmentation without compromising model performance, especially when the large, annotated dataset is available.

## 4.2 ENHANCED FEATURES FOR NOVEL DATASETS, CLASSES, AND TASKS

***Summary.*** The learned features manifest considerable generalizability and adaptability. The features can *direct inference* for organ segmentation on external datasets of CT volumes taken from different hospitals. The features can also be *fine-tuned* to segment novel organ/tumor classes and classify tumor sub-types with higher accuracy and less annotated data than those learned by self-supervision.

***Direct inference on external datasets.*** AI models trained on a specific dataset often encounter challenges in generalizing to novel datasets when a marked difference—referred to as a *domain gap*—exists between them (Zhang & Metaxas, 2023). While domain adaptation and generalization are prevalent research strategies to mitigate this challenge (Guan & Liu, 2021; Zhou et al., 2022a), we choose to address this issue by training a model on an expansive and diverse dataset (elaborated in Appendix A.1). We assume the domain gap between CT volumes from different hospitals is not as pronounced as those in computer vision. This is because of the relatively standardized nature of computer tomography as an imaging modality, where pixel intensity conveys consistent anatomical significance (Zhou et al., 2022b). AbdomenAtlas 1.1 presents impressive diversity, covering CT volumes with variations in contrast enhancement, reconstruction kernels, CT scanner types, and acquisition parameters. This breadth and diversity are imperative for developing an AI model with

Table 4: **Fine-tuning SuPreM on 66 novel classes.** Following the standard transfer learning paradigm, we fine-tune our SuPreM on the segmentation task of novel classes. These tasks include segmenting 19 muscles, 15 cardiac structures, 5 organs, and 24 vertebrae from TotalSegmentator, as well as three fine-grained pancreatic tumor types from the proprietary dataset. It is important to note that these classes were not part of the pre-training of SuPreM. We observe that SuPreM, supervised pre-trained on only a few classes, can transfer better than those self-supervised pre-trained on raw, unlabeled data measured by DSC scores (per-class results in Appendix D.3). In other words, it is the task of segmentation itself that can enhance the model's capability of segmenting novel-class objects. This benefit is much more straightforward and understandable than such self-supervised tasks as contextual prediction, mask image modeling, and instance discrimination in the context of transfer learning. We hypothesize that it is because the model learns to understand the concept of *objectness* in a broader sense through full supervision, as suggested by Kirillov et al. (2023), but this certainly deserves further exploration. In addition, an independent two-sample $t$-test was performed between the self-supervised pre-trained model and the supervised pre-trained model.

| novel class | self-super. | super. | $\Delta$ | novel class | self-super. | super. | $\Delta$ |
|---|---|---|---|---|---|---|---|
| humerus left | $92.8\pm0.7$ | $93.2\pm0.3$ [ns] | 0.4 | vertebrae L5 | $94.1\pm0.2$ | $95.7\pm0.3$ [****] | 1.6 |
| humerus right | $87.5\pm1.0$ | $95.0\pm0.5$ [****] | 7.6 | vertebrae L4 | $90.4\pm0.6$ | $93.0\pm0.5$ [****] | 2.6 |
| $\cdots$ (15 more classes) | | | | $\cdots$ (20 more classes) | | | |
| iliopsoas left | $84.4\pm0.3$ | $85.7\pm0.3$ [****] | 1.3 | vertebrae C2 | $86.8\pm2.0$ | $91.8\pm0.2$ [****] | 5.1 |
| iliopsoas right | $87.4\pm0.3$ | $88.7\pm0.2$ [****] | 1.3 | vertebrae C1 | $87.1\pm0.8$ | $87.4\pm0.8$ [ns] | 0.3 |
| **average (muscle)** | $93.9\pm0.1$ | $94.3\pm0.1$ [****] | 0.4 | **average (vertebrae)** | $86.4\pm0.3$ | $89.2\pm0.2$ [****] | 2.7 |
| | | | | | | | |
| trachea | $93.4\pm0.1$ | $93.4\pm0.1$ [ns] | 0.0 | | | | |
| heart myocardium | $88.9\pm0.2$ | $89.8\pm0.2$ [****] | 0.9 | | | | |
| $\cdots$ (11 more classes) | | | | PDAC | $53.3\pm0.4$ | $53.6\pm0.3$ [*] | 0.3 |
| urinary bladder | $90.5\pm0.9$ | $91.5\pm0.9$ [*] | 1.0 | Cyst | $41.5\pm0.3$ | $49.4\pm0.3$ [****] | 7.9 |
| pulmonary artery | $89.0\pm0.9$ | $92.0\pm0.2$ [****] | 3.0 | PanNet | $35.5\pm0.8$ | $46.0\pm0.5$ [****] | 10.5 |
| **average (cardiac)** | $88.9\pm0.1$ | $90.7\pm0.1$ [****] | 1.8 | **average (tumor)** | $43.4\pm0.3$ | $49.7\pm0.2$ [****] | 6.2 |

[ns] $P > 0.05$ [*] $P \leq 0.05$ [**] $P \leq 0.01$ [***] $P \leq 0.001$ [****] $P \leq 0.0001$

the robustness required to accommodate the variations present in novel datasets. We conduct external validation on several novel datasets sourced from Switzerland and East Asia to challenge the AI model on the data distribution that it has not encountered during the training. This result is referred to as *out-of-distribution* (OOD) performance. For comparison, we also collect the result achieved by dataset-specific AI models—those individually trained on the specific datasets—referred to as *independently and identically distributed* (IID) performance. As shown in Table 3, our SuPreM can be generalized well to novel data distribution without the need for further fine-tuning or adaptation, consistently offering OOD performance that matches or even exceeds that of its IID counterparts.

***Fine-tuning on novel classes.*** The value of transfer learning lies in fine-tuning the pre-trained models on novel scenarios (Zhou et al., 2021b), such as novel classes, image modalities, and vision tasks that are completely unseen during the pre-training. In this study, we evaluate the proficiency of SuPreM when transferred to a wide variety of novel classes for 3D image segmentation tasks[4]. These novel classes include 19 muscles, 15 cardiac structures, 5 organs, and 24 vertebrae from the TotalSegmentator dataset, as well as three fine-grained pancreatic tumor types from the proprietary dataset. As shown in Table 4, our SuPreM, supervised pre-trained on 25 classes, can transfer better to novel classes than those self-supervised models pre-trained on raw, unlabeled data. We find that the pretext task of segmentation itself can enhance the model capability of segmenting novel classes. The benefit of same-task transfer learning, i.e., segmentation as pretext and target tasks, is much more straightforward and understandable than other pretext tasks such as contextual prediction, mask image modeling, and instance discrimination. Through full supervision in segmentation tasks, the model learns to understand the concept of *objectness*[5], wherein the model gains a more profound understanding of what characterizes an object. The model does not just recognize predefined objects but begins to understand the foundational factors of objects in general. Such factors include texture, boundary, shape, size, and other low-level visual cues that are often deemed essential for image segmentation. This resonates with our assertion in the introduction: just as classification-based

---

[4]The fine-tuning performance of 17 seen classes, detailed in Appendix D.2, is promising, but this is expected because the model is exposed to more examples of these classes in both pre-training and fine-tuning phases.

[5]Objectness refers to the inherent attributes that distinguish something as an object within an image, differentiating it from the background or other entities.

(a) PDAC     (b) Cyst     (c) PanNET

Figure 3: **Fine-tuning SuPreM on fine-grained tumor classification.** We plot receiver operating characteristic (ROC) curves to evaluate the transfer learning performance of tumor classification. Detecting Cysts and PanNETs raises additional challenges for AI because these lesions exhibit a greater variety of texture patterns than PDACs. This diversity in texture patterns is reflected in the values of the Area Under the Curve (AUC) that we obtained. For all three sub-types of pancreatic tumors, SuPreM (in red) demonstrates superior performance over the self-supervised model (Tang et al., 2022) (in gray), showcasing its effectiveness in fine-grained tumor classification.

features from ImageNet transfer optimally for classification tasks (Huh et al., 2016; He et al., 2019; Zoph et al., 2020; Ridnik et al., 2021), segmentation-based features are optimal for segmentation tasks. Our findings do not negate the value of self-supervised pre-training. With 9,262 CT volumes, should self-supervised pre-training outperforms supervised pre-training in model transferability in the future, its value will be further highlighted by eliminating the need for manual annotations.

***Fine-tuning on novel tasks.*** We have investigated the cross-task transfer learning ability of SuPreM between organ segmentation and fine-grained tumor classification. The distance between the two tasks is much larger than transferring among segmentation tasks. It is challenging to benchmark fine-grained tumor classification, particularly due to the scarcity of annotations in public datasets (often limited to hundreds of tumors). To overcome this limitation, we employed our proprietary dataset (Xia et al., 2022), which comprises 3,577 annotated pancreatic tumors, including detailed sub-types: 1,704 PDACs, 945 Cysts, and 928 PanNETs. This extensive dataset enabled us to thoroughly assess the transfer learning ability of SuPreM in tumor-related tasks. Figure 3 shows that supervised models (SuPreM) transfer better to target classification tasks than self-supervised models (Tang et al., 2022), leading to improved Area Under the Curve (AUC) for identifying each tumor type. Notably, the transfer learning results detailed in Appendix D.4 reveal a sensitivity of 86.1% and specificity of 95.4% for PDAC detection. This performance surpasses the average radiologist's performance in PDAC identification by 27.6% in sensitivity and 4.4% in specificity, as reported in Cao et al. (2023). Moreover, Appendix Figure 8 shows that SuPreM requires 50% fewer manual annotations for fine-grained tumor classification than self-supervised pre-training. This is particularly critical for tumor imaging tasks because annotating tumors requires much more effort and often relies on the availability of pathology reports.

## 5    CONCLUSION AND DISCUSSION

This study examines the transfer learning ability of supervised models that are pre-trained on 3D annotated datasets and fine-tuned on 3D image segmentation tasks. We start by constructing AbdomenAtlas 1.1, an extensive collection of **9,262** three-dimensional CT volumes with high-quality, per-voxel annotations. The magnitude of this dataset is unprecedented regarding data volume (**2,789,975 images**), granularity of annotations (**251,323 masks**), and inclusive diversity (**88 hospitals**). This dataset facilitates the development of a suite of pre-trained models, termed SuPreM, that can be effectively transferred to a broad spectrum of 3D image segmentation tasks. Notably, SuPreM transfers better than all existing 3D models by a large margin, especially when transferred to under-annotated datasets. The model trained with 21 CT volumes, 672 masks, and 40 GPU hours shows a transfer learning ability similar to that trained with 5,050 CT volumes and 1,152 GPU hours, highlighting the remarkable efficiency of supervised pre-training. We also demonstrate that the learned features can *direct inference* effectively on external datasets and *fine-tune* to segment novel classes and classify multiple types of tumors with higher accuracy and less annotated data than those learned by self-supervision.

ACKNOWLEDGMENTS

This work was supported by the Lustgarten Foundation for Pancreatic Cancer Research and the Patrick J. McGovern Foundation Award. This work has utilized the GPUs provided partially by ASU Research Computing and NVIDIA. We appreciate the effort of the MONAI Team to provide open-source code for the community. We thank Chongyu Qu, Yixiong Chen, Junfei Xiao, Jie Liu, Yucheng Tang, Tiezheng Zhang, Yaoyao Liu, Chen Wei, Fengrui Tian, Yu-Cheng Chou, Angtian Wang, and Dora Zhiyu Yang for their constructive suggestions at several stages of the project.

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
