# OpenReview forum: "How Well Do Supervised 3D Models Transfer to Medical Imaging Tasks?"
_ICLR.cc/2024/Conference — ICLR 2024 oral_

### Official Review · Reviewer_osvs · 2023-10-28

**Soundness:** 3 good
**Presentation:** 3 good
**Contribution:** 3 good
**Rating:** 6
**Confidence:** 3

**Summary:**

The paper introduces a novel dataset (IMAGENETCT-9K) containing 9,262 CT volumes along with their respective voxel-level masks. Furthermore, the paper pretrains various models on this dataset and fine-tunes it on other publicly available benchmarks, achieving SOTA performance.

**Strengths:**

The dataset appears to be comprehensive and holds great promise. I believe that making this dataset and the pretrained weights publicly available can contribute to advancements in the field.

**Weaknesses:**

The weakness could relate to the details of the pretraining strategy. Typically, image-wise [1, 2] or pixel-wise [3] pre-training relies on the InfoNCE loss for clustering embedding samples in the latent space, rather than directly applying penalties based on labels via cross-entropy loss. It would be more interesting to observe results achieved through category-guided InfoNCE loss [4, 5] pre-training using this dataset.

Additionally, there exist many promising domain transfer methods, yet the paper appears to lack exploration in this area. The current approach appears to be straightforward fine-tuning on other datasets, such as TotalSegmentator and JHH.

[1] Self-training with Noisy Student improves ImageNet classification

[2] Unsupervised Learning of Visual Features by Contrasting Cluster Assignments

[3] Dense Contrastive Learning for Self-Supervised Visual Pre-Training

[4] Supervised Contrastive Learning

[5] Exploring Cross-Image Pixel Contrast for Semantic Segmentation

**Questions:**

I don't have a lot questions since the paper's primary contribution lies in the dataset. I would like to confirm whether the dataset and the pre-trained weights will be made accessible to the public.

**Details Of Ethics Concerns:**

I missed the data privacy of the patients throughout the processes of data collection, storage, and sharing in the paper. I have observed a 'pending' status in Table 5 of your Appendix A and I believe it is essential for the authors to address this issue appropriately.

---

> ### Comment · Reviewer_osvs · 2023-11-22
>
> My questions are well solved.

---

> ### Author Response · Authors · 2023-11-22
> **Response to Reviewer osvs (1/2)**
>
> Thank you for acknowledging our general response. We appreciate your thoughtful feedback and accolades on our contribution: *“the dataset appears to be comprehensive and holds great promise…making this dataset and the pretrained weights publicly available can contribute to advancements in the field.”*
>
> ---
>
> > **Q1.** The weakness could relate to the details of the pretraining strategy. Typically, image-wise [1, 2] or pixel-wise [3] pre-training relies on the InfoNCE loss for clustering embedding samples in the latent space, rather than directly applying penalties based on labels via cross-entropy loss. It would be more interesting to observe results achieved through category-guided InfoNCE loss [4, 5] pre-training using this dataset.
>
> Thanks for your suggestion. We must admit that category-guided InfoNCE loss was not our first try by default because it is not a standard approach for medical segmentation tasks as of now. [[Ma et al., MEDIA 2021](https://www.sciencedirect.com/science/article/pii/S1361841521000815)] have listed all the popular losses for medical segmentation tasks, each of these loss functions may have their own strengths to specific scenarios, such as dealing with unbalanced class. Therefore, we initially selected a combination of Dice loss and binary cross-entropy loss, the most commonly used segmentation loss, as recommended by the introduction and Figure 1 in [[Ma et al., MEDIA 2021](https://www.sciencedirect.com/science/article/pii/S1361841521000815)].
>
> But we are very much interested in exploring more loss options. Category-guided InfoNCE loss, as you suggested, is potentially good because it can cluster embedding samples in the latent space and has proven effective for supervised learning in natural image classification [4] and segmentation tasks [5]. Thanks for sharing the references. The references [1, 2, 3], however, discussed self-supervised learning, so their applicability and effectiveness in the context of supervised learning for segmentation tasks might not have been thoroughly explored. Following your suggestion, a comprehensive comparison between InfoNCE loss and our dice + bce loss will be included in our final version.
>
> ---
>
> > **Q2.** Additionally, there exist many promising domain transfer methods, yet the paper appears to lack exploration in this area. The current approach appears to be straightforward fine-tuning on other datasets, such as TotalSegmentator and JHH.
>
> The current approach is pretty robust (evidenced in Table 3) because our dataset covers a variety of domains (i.e., 68 hospitals with different scanners and protocols). Therefore, models pre-trained on this dataset are expected to be generalizable for novel domains, e.g., TotalSegmentator, FLARE’23, and JHH. These three datasets are completely unseen during the pre-training and represent a variety of diversity. Specifically, the TotalSegmentator dataset represents for the Central European population from Switzerland, the FLARE’23 dataset represents for East Asian population from China, and the JHH dataset represents for another population (anonymous for peer review). Our models achieve comparable or even superior performance to the IID counterparts (Table 3). Therefore, domain transfer becomes less important if the model is pre-trained on large and diverse datasets (elaborated in the next two points).
>
> 1. The domain transfer problem could be solved by methodology innovation, as you suggested, and also by training AI models on enormous datasets. This point has been more clear recently demonstrated by large language models (GPT) and vision foundation models (SAM), which show incredible performance in “new domain”. However, this achievement may not be directly attributed to method-driven solutions for domain transfer, but simply because the AI might have been trained on similar sentences or images. This was also pointed out by Yann Lecun—*“[beware of testing on the training set](https://twitter.com/ylecun/status/1723752958037315874)”*—in response to the incredible results achieved by GPT.
>
> 2. In some sense, our paper explores dataset-driven solutions for domain transfer. The robust performance of our models when direct inference on multiple domains could also be attributed to our large-scale, fully-annotated medical dataset—as one of our major contributions. The release of this dataset can foster AI models that are more robust than the majority of existing models that are only trained on a few hundred CT volumes from limited domains. We completely agree that existing domain transfer methods could be supplemented with direct inference and fine-tuning to further improve AI performance.
>
> ---
>
> **Reference**
>
> - Ma, Jun, Jianan Chen, Matthew Ng, Rui Huang, Yu Li, Chen Li, Xiaoping Yang, and Anne L. Martel. "Loss odyssey in medical image segmentation." *Medical Image Analysis* (2021).

---

> ### Author Response · Authors · 2023-11-22
> **Response to Reviewer osvs (2/2)**
>
> > **Q3.** I would like to confirm whether the dataset and the pre-trained weights will be made accessible to the public.
>
> We confirm that the dataset, the pre-trained weights, and the source code will be made accessible to the public upon the acceptance of the paper. Pre-trained weights have already been pre-leased for peer review and public audience through the links in the general responses. Feel free to test it out. And we have already attached our source code to the supplementary material. The dataset, however, cannot be pre-leased because we are in the process of licensing our ImageNetCT-9K.
>
> ---
>
> > **Q4.** I missed the data privacy of the patients throughout the processes of data collection, storage, and sharing in the paper. I have observed a 'pending' status in Table 5 of your Appendix A and I believe it is essential for the authors to address this issue appropriately.
>
> Ensuring the data privacy of the patients is our top priority. We collected over 9K CT volumes from publicly available datasets (Table 1 and Appendix B.1 Table 5) and annotated 257K organ/tumor masks in addition to what public datasets already provided. The total size of our ImageNetCT-9K is around 500 GB. We commit to releasing the entire dataset to the public upon the acceptance of the paper. Currently, ImageNetCT-9K is in the process of acquiring a proper license, so we put a “pending” status for now.

---

### Official Review · Reviewer_mSzX · 2023-11-01

**Soundness:** 3 good
**Presentation:** 3 good
**Contribution:** 3 good
**Rating:** 8
**Confidence:** 3

**Summary:**

The paper investigates the transfer learning ability of self-supervised and fully-supervised foundational models for medical image segmentation. For this purpose, the authors collect a very large, labeled 3D CT scans to be used for pre-trainining. The results show that supervised models have better transfer learning ability compared to self-supervised counterparts, by also saving a significant GPU time. Moreover, since the collected data is very diverse and large, it generalizes well to OOD dataset even surpasses the performance of the models trained on the OOD datasets.

**Strengths:**

- Lack of large annotated datasets is a huge problem in medical imaging due to the cost of collecting labelled data. Publicly available pretrained models on such large datasets are huge assets for medical imaging.

- The paper presents extensive experiments showing the benefit of supervised pre-training compared to the unsupervised counterparts.

**Weaknesses:**

- UniverSeg [1] is another paper that trains networks on a very large dataset, 22K scans, which is even larger than this paper. Although Arxiv version of UniverSeg is available since April 2023, I see this work and UniverSeg as concurrent works since UniverSeg is recently presented in ICCV. However, I still think that mentioning UniverSeg and discussing the similarities/differences in the final version would be useful.

[1] https://universeg.csail.mit.edu/

- How do the models trained on the collected large CT dataset generalize to novel modalities such as MRI?

**Questions:**

Please address my concerns in the weaknesses section.

---

> ### Author Response · Authors · 2023-11-22
> **Response to Reviewer mSzX**
>
> We appreciate the acknowledgment of the scientific impact of our ImageNetCT-9K as *“lack of large annotated datasets is a huge problem in medical imaging…”*. Thank you so much for praising our pre-trained models as *“huge assets for medical imaging”*.
>
> ---
>
> > **Q1.** UniverSeg [1] is another paper that trains networks on a very large dataset, 22K scans, which is even larger than this paper. Although Arxiv version of UniverSeg is available since April 2023, I see this work and UniverSeg as concurrent works since UniverSeg is recently presented in ICCV. However, I still think that mentioning UniverSeg and discussing the similarities/differences in the final version would be useful.
>
> Thank you for sharing UniverSeg with us—its relevance to our work is significant and appreciated. We have now included a discussion of UniverSeg, along with other concurrent and related works, in the revised Section 2. These works share a level of **similarity** with ours, aiming to develop medical AI models that can segment multiple anatomical structures and generalize to out-of-distribution datasets.
>
> **Opinions on UniverSeg.** UniverSeg demonstrates an impressive ability to generalize across various unseen tasks and modalities without the need for fine-tuning, offering computational efficiency. However, its current performance falls short of the benchmarks set by fine-tuned models (as their upper bound reference). **In contrast**, our research aims to advance the fine-tuning performance by pre-training models on larger, per-voxel annotated datasets. We believe that our ImageNetCT-9K dataset could also further enhance UniverSeg's generalization ability.
>
> **Perspectives on additional related and future work** The forthcoming public release of ImageNetCT-9K and SuPreM is anticipated to positively influence Segment Anything Models (SAM) within the medical field. We hold the same belief with [[Kirillov et al. ICCV 2023](https://openaccess.thecvf.com/content/ICCV2023/papers/Kirillov_Segment_Anything_ICCV_2023_paper.pdf)] that the foundation models for image segmentation require supervised training on board data at scale, and we both contribute to a large-scale annotated dataset in the respective field. In addition, we already have some proof of concept that the model tends to understand **objectness** in a broader sense through full supervision in Table 4.
>
> ---
>
> > **Q2.** How do the models trained on the collected large CT dataset generalize to novel modalities such as MRI?
>
> This is a very good point. We think transfer learning across different imaging modalities, such as from CT to MRI, might be less effective compared to transfers within the same modality, primarily due to the significant differences in their imaging techniques. The discrepancies in image acquisition methods between CT and MRI result in distinct intensity values and ranges. Nonetheless, our pre-trained model could still be valuable for abdominal MRI applications. This is because the underlying anatomical structures remain consistent across both CT and MRI, allowing for the potential transfer of shared knowledge.
>
> Given the constraints of time, we are unable to include this specific experiment in the rebuttal period. However, we plan to incorporate a study focused on **abdominal MRI tasks** in the final version of our paper. With the release of our ImageNetCT-9K and SuPreM, we look forward to promoting a collaborative effort to thoroughly evaluate the capabilities of transfer learning. This includes exploring a wider range of medical modalities, such as T1, T1c, T2, Flair, and Ultrasound, and extending into general 3D vision tasks involving diverse data formats like point clouds, voxel occupancy grids, meshes, and implicit surface models (e.g., signed distance functions). We have now included in the future work section and Appendix F.4.
>
> ---
>
> **Reference**
> - Kirillov, Alexander, Eric Mintun, Nikhila Ravi, Hanzi Mao, Chloe Rolland, Laura Gustafson, Tete Xiao et al. "Segment anything." *ICCV* (2023).

---

> > ### Comment · Reviewer_mSzX · 2023-11-22
> > **Thanks**
> >
> > Thanks for the well-prepared rebuttal. I am still suggesting acceptance of this paper.

---

### Official Review · Reviewer_bY7Q · 2023-11-02

**Soundness:** 3 good
**Presentation:** 3 good
**Contribution:** 3 good
**Rating:** 6
**Confidence:** 5

**Summary:**

The authors collected publicly available and private CT data, and obtained over 9000 CT data cases by manually correcting pseudo-labels. These data can support the segmentation of 32 organs and a small number of tumors. Through experiments, it was discovered that the supervised pre-training method outperforms self-supervised pre-training and supervised pre-training with fewer samples.

**Strengths:**

The authors collected over 9000 cases of publicly available and private 3D CT datasets, and manually corrected annotation errors, making it the largest dataset for multi-organ segmentation currently available.

**Weaknesses:**

1. The paper explores the transferability of supervised learning by combining multiple 3D CT segmentation datasets. Similar work has been done in various fields, such as training various tasks and data in computer vision, which has shown improved results across tasks. This paper incorporates 3D CT data and tasks, with the only difference being the collection of more publicly available and private data, without bringing new insights or technological innovations to the community.
2. The conclusion that supervised pre-training has an advantage over other pre-training methods in 3D medical imaging is generalized to general 3D vision tasks in an inconsistent manner. The introduction discusses pre-training strategies for 3D vision tasks, but the experiments are all conducted on medical images. It is well known that there are significant differences between medical images and natural images, and whether the experimental conclusions on 3D CT data can be extended to other 3D vision tasks is not explored in the paper.
3. The fused dataset of over 9000 CT cases collected by the authors includes segmentation of 32 organs and tumors, but the evaluation in the experimental section focuses more on organs, lacking an evaluation of tumor segmentation performance. Comparatively, organ segmentation is less challenging in terms of generalization, while tumor segmentation is more complex. In the external dataset, more focus should be given to tumor segmentation, as it is more susceptible to a series of generalization issues caused by differences in populations, devices, and diseases in practical application scenarios.
4. Unfair comparison in the experimental section is a fatal flaw. Although the authors claim that collecting 9000 data is their contribution, the same 9000 data were not used for pre-training when comparing with other self-supervised/supervised pre-training methods. Therefore, it is not rigorous to conclude that SPT is superior to other supervised/self-supervised methods.
5. The results of fine-tuning SPT on 63 novel classes are not impressive. Although there is no comparison with totalsegmentator, the performance of totalsegmentator trained on 1000 data seems to surpass what is reported in Table 4. For example, totalsegmentator can achieve over 95% Dice on iliopsoas, while it is less than 90% in the paper.

**Questions:**

1. The paper does not clearly explain how the three expert radiologists collaborated to clean the data, including how they worked together, established uniform standards, and how they corrected tumor masks in the pseudo labels. What was the time cost involved, and so on?
2. The "novel datasets" claimed in Table 3 is inappropriate. It should be referred to as the external dataset.

**Details Of Ethics Concerns:**

This paper uses internal medical data for training, so it may need ethical approval for use.

---

> ### Author Response · Authors · 2023-11-22
> **Response to Reviewer bY7Q (1/6)**
>
> We would like to thank you for your diligent efforts and constructive suggestions on our paper, which have helped us think more deeply. In the following, we have provided a point-by-point response to all questions raised.
>
> ---
>
> > **Q1.** The paper explores the transferability of supervised learning by combining multiple 3D CT segmentation datasets. Similar work has been done in various fields, such as training various tasks and data in computer vision, which has shown improved results across tasks. This paper incorporates 3D CT data and tasks, with the only difference being the collection of more publicly available and private data, without bringing new insights or technological innovations to the community.
>
> Creating large-scale (9K) per-voxel annotated medical datasets of over 25 classes and making this resource publicly available takes a village. Your acknowledgment of our dataset as *“the largest dataset for multi-organ segmentation currently available”* is greatly appreciated. While it is common in computer vision to improve performance through joint training on a combination of existing datasets, our study overcomes technical barriers and brings new insights as follows.
>
> 1. ImageNetCT-9K is NOT a simple combination of existing datasets. We have now included Appendix B.1 Figure 5, in conjunction with Table 5, to better illustrate the evolution from public datasets to our ImageNetCT-9K. The 9K CT volumes in the combination of public datasets only contain a total of **39K** annotated organ masks, while our ImageNetCT-9K provides **296K** annotated organ/tumor masks for these CT volumes, substantially increasing the number of masks by **7.6** times.
>
> 2. Creating **296K** high-quality organ/tumor masks for 9K CT volumes requires extensive medical knowledge and annotation cost (much more difficult than annotating natural images). Based on our experience and those reported in [[Park et al., Diagnostic and interventional imaging 2020](https://pubmed.ncbi.nlm.nih.gov/31358460/)], trained radiologists annotate abdominal organs at a rate of 30–60 minutes per organ per three-dimensional CT volume. This translates to **247K** human hours for completing ImageNetCT-9K. We employed a highly efficient annotation method, combining AI with the expertise of three radiologists using active learning (details in **Q6**), to overcome this challenge and produce the largest annotated dataset to date.
>
> 3. ImageNetCT-9K can be used as a training resource and a testbed for AI algorithms. As acknowledged by Reviewer fMfJ, it can *“have a good impact on the whole community.”* Our study introduces **new insights** on the use of ImageNetCT-9K in transfer learning. We delve into a key debate within general computer vision: the comparative effectiveness of representation learning via human-annotated data (supervised pre-training) versus raw data (self-supervised pre-training). As Reviewer fMfJ acknowledged, *”this paper concluded on the debate of whether self-supervised or supervised pre-training lead to better performance and data efficiency. This debate **had not be resolved** without the invention of a fully-annotation dataset of such scale.”* We are committed to making ImageNetCT-9K publicly available, thereby enabling further research to derive insights from various angles and foster technological innovations. Reviewer mSzX recognized that *”publicly available pretrained models on such large datasets are huge assets for medical imaging.”*
>
>
> ---
>
> **Reference**
>
> - Park, S., L. C. Chu, E. K. Fishman, A. L. Yuille, B. Vogelstein, K. W. Kinzler, K. M. Horton et al. "Annotated normal CT data of the abdomen for deep learning: Challenges and strategies for implementation." *Diagnostic and interventional imaging* (2020).

---

> ### Author Response · Authors · 2023-11-22
> **Response to Reviewer bY7Q (2/6)**
>
> > **Q2.** The conclusion that supervised pre-training has an advantage over other pre-training methods in 3D medical imaging is generalized to general 3D vision tasks in an inconsistent manner. The introduction discusses pre-training strategies for 3D vision tasks, but the experiments are all conducted on medical images. It is well known that there are significant differences between medical images and natural images, and whether the experimental conclusions on 3D CT data can be extended to other 3D vision tasks is not explored in the paper.
>
> We completely agree with your comments and have now revised our title, abstract, introduction, and conclusion to narrow down our current research scope to 3D medical imaging.
>
> The potential applications and implications of our dataset have now been included in the future work section. Given that our dataset includes detailed per-voxel annotations for 25 organs and tumors, it enables the automatic generation of 3D shape representations. These representations can be formatted as point clouds, voxel occupancy grids, meshes, and implicit surface models (e.g., signed distance functions), each catering to different algorithmic needs. We anticipate our dataset could be useful for a variety of other 3D medical vision tasks [[Li et al., 2023](https://arxiv.org/abs/2308.16139)], such as pose estimation, surface reconstruction, depth estimation, etc.
>
> ---
>
> > **Q3.** The fused dataset of over 9000 CT cases collected by the authors includes segmentation of 32 organs and tumors, but the evaluation in the experimental section focuses more on organs, lacking an evaluation of tumor segmentation performance. Comparatively, organ segmentation is less challenging in terms of generalization, while tumor segmentation is more complex. In the external dataset, more focus should be given to tumor segmentation, as it is more susceptible to a series of generalization issues caused by differences in populations, devices, and diseases in practical application scenarios.
>
> **Tumor-related tasks** were evaluated in Figure 1. Due to space constraints and the breadth of our experiments, we initially averaged the performance metrics for both tumor and organ segmentation in Figure 1, for which we apologize if this led to any confusion. We completely agree with you that evaluating tumor-related tasks is much more significant and challenging than organ tasks. We have now explicitly reported tumor segmentation performance in Table 4 and tumor classification performance in Figure 3.
>
> We would like to stress the challenges in benchmarking tumor segmentation/classification, particularly due to the scarcity of annotations in publicly available datasets (often limited to hundreds of tumors). To overcome this limitation, we employed our proprietary dataset, which comprises **3,577** annotated pancreatic tumors, including detailed sub-types: **1,704 PDACs**, **945 Cysts**, and **928 PanNets**. This extensive dataset enabled us to thoroughly assess the transfer learning ability of our pre-trained models in tumor-related tasks. Notably, the transfer learning results detailed in Appendix E.4 Figure 13 demonstrate a sensitivity of 86.1% and specificity of 95.4% for PDAC detection. This performance surpasses the average radiologist's performance in PDAC identification by 27.6% in sensitivity and 4.4% in specificity, as reported in [[Cao et al., Nat Med 2023](https://www.nature.com/articles/s41591-023-02640-w)]. This is one of the demonstration how our pre-trained models can be deployed for clinical applications. If we have an honor to present this work at ICLR, we can perhaps elaborate more.
>
> **Generalization issues.** Thanks for bringing this into our attention. Through our experiment, we have shown the generalizability in terms of populations in Table 3 (i.e. TotalSegmentator (representing the Central European population from Switzerland) and FLARE’23 (the East Asian population from China)). Moreover, our proprietary dataset contains CT scans taken by a variety of vendors, e.g. Siemens, GE, Philips, and Toshiba, as well as scanners 16-/64-slice MDCT and Dual-source MDCT. The promising results in all three external datasets—with various populations, devices, and diseases—suggest the clinical impact of our pre-trained models in practical application scenarios.
>
> ---
>
> **Reference**
>
> - Li, Jianning, Antonio Pepe, Christina Gsaxner, Gijs Luijten, Yuan Jin, Narmada Ambigapathy, Enrico Nasca et al. "MedShapeNet--A Large-Scale Dataset of 3D Medical Shapes for Computer Vision." *arXiv* (2023).
> - Cao, Kai, Yingda Xia, Jiawen Yao et al. Large-scale pancreatic cancer detection via non-contrast CT and deep learning. *Nature Medicine* (2023).

---

> ### Author Response · Authors · 2023-11-22
> **Response to Reviewer bY7Q (3/6)**
>
> > **Q4.** Unfair comparison in the experimental section is a fatal flaw. Although the authors claim that collecting 9000 data is their contribution, the same 9000 data were not used for pre-training when compared with other self-supervised/supervised pre-training methods. Therefore, it is not rigorous to conclude that SPT is superior to other supervised/self-supervised methods.
>
> Thank you for your valuable feedback. Ensuring a fair and rigorous comparison is our top priority. Therefore, we must clarify that while we have developed and released a suite of models pre-trained on 9K data, these models were NOT used for comparisons against other self-supervised/supervised pre-training methods within this paper. The only experiment that used 9K-models was the direct inference on external datasets (Table 3), where we benchmarked against methods trained specifically on those datasets, representing an upper bound in performance. This benchmark is to ensure that the released models are the most effective and robust ones (as we could provide) for the research community to directly use.
>
> For the other experiments, our aim was to evaluate the efficacy of supervised pre-training relative to other pre-training methods, so we designed them within a controlled setting.
>
> 1. For supervised pre-training, the largest study to date was by [[Liu et al., ICCV 2023](https://openaccess.thecvf.com/content/ICCV2023/papers/Liu_CLIP-Driven_Universal_Model_for_Organ_Segmentation_and_Tumor_Detection_ICCV_2023_paper.pdf)], which was developed on 3,410 (2,100 for training and 1,310 for validation) annotated CT volumes. For self-supervised pre-training, the largest one was by [[Tang et al., CVPR 2022](https://openaccess.thecvf.com/content/CVPR2022/papers/Tang_Self-Supervised_Pre-Training_of_Swin_Transformers_for_3D_Medical_Image_Analysis_CVPR_2022_paper.pdf)], which was trained on 5,050 unannotated CT volumes. To do a rigorous comparison, we **benchmarked** with these advanced pre-training methods by pre-training our model using 2,100 CT volumes (*same as Liu et al. and fewer than Tang et al.*) in the Table 2, Figure 1 and Appendix C.2 Figure 8.
> 2. We further **scaled down** the number of CT volumes to 21 to explore the edge of our supervised pre-training method. Surprisingly, Figure 2a shows these experiments and demonstrates that the model trained with 21 CT volumes, 672 masks, and 40 GPU hours shows a transfer learning ability similar to [[Tang et al., CVPR 2022](https://openaccess.thecvf.com/content/CVPR2022/papers/Tang_Self-Supervised_Pre-Training_of_Swin_Transformers_for_3D_Medical_Image_Analysis_CVPR_2022_paper.pdf)] trained with 5,050 CT volumes and 1,152 GPU hours.
> 3. Lastly, we **scaled up** the SOTA self-supervised method [[Tang et al., CVPR 2022](https://openaccess.thecvf.com/content/CVPR2022/papers/Tang_Self-Supervised_Pre-Training_of_Swin_Transformers_for_3D_Medical_Image_Analysis_CVPR_2022_paper.pdf)] by pre-training it on the same 9K CT volumes. Under this similar setting, our supervised pre-training 9K model substantially outperforms the SOTA self-supervised 9K-model (see Appendix C2 Table 8).
>
> ---
>
> **Reference**
>
> - Liu, Jie, Yixiao Zhang, Jie-Neng Chen, Junfei Xiao, Yongyi Lu, Bennett A Landman, Yixuan Yuan, Alan Yuille, Yucheng Tang, and Zongwei Zhou. "Clip-driven universal model for organ segmentation and tumor detection." *ICCV* (2023).
> - Tang, Yucheng, Dong Yang, Wenqi Li, Holger R. Roth, Bennett Landman, Daguang Xu, Vishwesh Nath, and Ali Hatamizadeh. "Self-supervised pre-training of swin transformers for 3d medical image analysis." *CVPR* (2022).

---

> ### Author Response · Authors · 2023-11-22
> **Response to Reviewer bY7Q (4/6)**
>
> For your convenience, we have summarized all the models we compared in Appendix C.2 Table 8. A concise version is provided as follows, where the name with a star (*) denotes it is **implemented and pre-trained** by us.
>
> - **Self-supervised pre-training**
>
> | name &nbsp; &nbsp; &nbsp; &nbsp; &nbsp;&nbsp; &nbsp; &nbsp; &nbsp; &nbsp;| backbone &nbsp; &nbsp; &nbsp; &nbsp; &nbsp;| params &nbsp; &nbsp; &nbsp; &nbsp; &nbsp;| pre-trained data &nbsp; &nbsp; &nbsp; &nbsp; &nbsp;| paper &nbsp; &nbsp; &nbsp; &nbsp; &nbsp;|
> |  ----  | ----  |  ----  |  ----  |  ----  |
> | Models Genesis | U-Net | 19.08M | 623  | [Zhou et al.](http://www.cs.toronto.edu/~liang/Publications/ModelsGenesis/MICCAI_2019_Full.pdf) |
> | UniMiSS | U-Net | 19.08M | 5,022 | [Xie et al.](https://link.springer.com/chapter/10.1007/978-3-031-19803-8_33) |
> | NV | Swin UNETR | 62.19M | 5,050  | [Tang et al.](https://openaccess.thecvf.com/content/CVPR2022/papers/Tang_Self-Supervised_Pre-Training_of_Swin_Transformers_for_3D_Medical_Image_Analysis_CVPR_2022_paper.pdf) |
> | NV* | Swin UNETR | 62.19M | 1,000  | [Tang et al.](https://openaccess.thecvf.com/content/CVPR2022/papers/Tang_Self-Supervised_Pre-Training_of_Swin_Transformers_for_3D_Medical_Image_Analysis_CVPR_2022_paper.pdf) |
> | NV* | Swin UNETR | 62.19M | 3,000  | [Tang et al.](https://openaccess.thecvf.com/content/CVPR2022/papers/Tang_Self-Supervised_Pre-Training_of_Swin_Transformers_for_3D_Medical_Image_Analysis_CVPR_2022_paper.pdf) |
> | NV* | Swin UNETR | 62.19M | 5,050  | [Tang et al.](https://openaccess.thecvf.com/content/CVPR2022/papers/Tang_Self-Supervised_Pre-Training_of_Swin_Transformers_for_3D_Medical_Image_Analysis_CVPR_2022_paper.pdf) |
> | NV* | Swin UNETR | 62.19M | 9,262  | [Tang et al.](https://openaccess.thecvf.com/content/CVPR2022/papers/Tang_Self-Supervised_Pre-Training_of_Swin_Transformers_for_3D_Medical_Image_Analysis_CVPR_2022_paper.pdf) |
>
> - **Supervised pre-training**
>
> | name &nbsp; &nbsp; &nbsp; &nbsp; &nbsp;&nbsp; &nbsp; &nbsp; &nbsp; &nbsp;| backbone &nbsp; &nbsp; &nbsp; &nbsp; &nbsp;| params &nbsp; &nbsp; &nbsp; &nbsp; &nbsp;| pre-trained data &nbsp; &nbsp; &nbsp; &nbsp; &nbsp;| paper &nbsp; &nbsp; &nbsp; &nbsp; &nbsp;|
> |  ----  | ----  |  ----  |  ----  |  ----  |
> | Med3D | U-Net | 19.08M | 1,638 | [Chen et al.](https://arxiv.org/pdf/1904.00625.pdf) |
> | DoDNet | U-Net | 19.08M | 920 | [Zhang et al.](https://openaccess.thecvf.com/content/CVPR2021/papers/Zhang_DoDNet_Learning_To_Segment_Multi-Organ_and_Tumors_From_Multiple_Partially_CVPR_2021_paper.pdf)  |
> | DoDNet* | U-Net | 19.08M | 920 | [Zhang et al.](https://openaccess.thecvf.com/content/CVPR2021/papers/Zhang_DoDNet_Learning_To_Segment_Multi-Organ_and_Tumors_From_Multiple_Partially_CVPR_2021_paper.pdf)  |
> | Universal Model | U-Net | 19.08M | 3,410 | [Liu et al.](https://openaccess.thecvf.com/content/ICCV2023/papers/Liu_CLIP-Driven_Universal_Model_for_Organ_Segmentation_and_Tumor_Detection_ICCV_2023_paper.pdf)  |
> | Universal Model | Swin UNETR | 62.19M | 3,410 | [Liu et al.](https://openaccess.thecvf.com/content/ICCV2023/papers/Liu_CLIP-Driven_Universal_Model_for_Organ_Segmentation_and_Tumor_Detection_ICCV_2023_paper.pdf)  |
> | SuPreM* | U-Net | 19.08M | 9,262  | ours |
> | SuPreM* | Swin UNETR | 62.19M | 9,262  | ours |
> | SuPreM* | Swin UNETR | 62.19M | 21  | ours |
> | SuPreM* | Swin UNETR | 62.19M | 2,100  | ours |
> | SuPreM* | SegResNet | 470.13M | 9,262  | ours |
>
> ---
>
> **Reference**
>
> - Tang, Yucheng, Dong Yang, Wenqi Li, Holger R. Roth, Bennett Landman, Daguang Xu, Vishwesh Nath, and Ali Hatamizadeh. "Self-supervised pre-training of swin transformers for 3d medical image analysis." *CVPR* (2022).
> - Zhou, Zongwei, Vatsal Sodha, Md Mahfuzur Rahman Siddiquee, Ruibin Feng, Nima Tajbakhsh, Michael B. Gotway, and Jianming Liang. "Models genesis: Generic autodidactic models for 3d medical image analysis." *MICCAI* (2019).
> - Xie, Yutong, Jianpeng Zhang, Yong Xia, and Qi Wu. "Unimiss: Universal medical self-supervised learning via breaking dimensionality barrier." *ECCV* (2022).
> - Chen, Sihong, Kai Ma, and Yefeng Zheng. "Med3d: Transfer learning for 3d medical image analysis." *arXiv* (2019).
> - Zhang, Jianpeng, Yutong Xie, Yong Xia, and Chunhua Shen. "DoDNet: Learning to segment multi-organ and tumors from multiple partially labeled datasets." *CVPR* (2021).
> - Liu, Jie, Yixiao Zhang, Jie-Neng Chen, Junfei Xiao, Yongyi Lu, Bennett A Landman, Yixuan Yuan, Alan Yuille, Yucheng Tang, and Zongwei Zhou. "Clip-driven universal model for organ segmentation and tumor detection." *ICCV* (2023).

---

> ### Author Response · Authors · 2023-11-22
> **Response to Reviewer bY7Q (5/6)**
>
> > **Q5.** The results of fine-tuning SPT on 63 novel classes are not impressive. Although there is no comparison with totalsegmentator, the performance of totalsegmentator trained on 1000 data seems to surpass what is reported in Table 4. For example, totalsegmentator can achieve over 95% Dice on iliopsoas, while it is less than 90% in the paper.
>
> In TotalSegmentator, the labels were largely generated by a **single** nnU-Net re-trained continually (see Figure 1b in [Wasserthal et al.](https://pubs.rsna.org/doi/full/10.1148/ryai.230024)). Depending solely on nnU-Net could introduce a potential label bias favoring the nnU-Net architecture. This means two points.
>
> 1. Their ground truth (revised pseudo labels) could be biased to the nnU-Net architecture. nnU-Net trained and tested on this dataset will achieve an **unreachable** performance as shown in Totalsegmentator [github](https://github.com/wasserth/TotalSegmentator/blob/master/resources/evaluate_results.txt). For example, they report the DSC score of 0.894 for pancreas segmentation, this number has never been reached literature. The MSD top 1 result (**[0.828](https://decathlon-10.grand-challenge.org/evaluation/challenge/leaderboard/)**); TCIA Pancreas-CT Dataset top 1 result (**[0.845](https://paperswithcode.com/sota/pancreas-segmentation-on-tcia-pancreas-ct)**); even the FELIX Project (producing the largest pancreas dataset in USA) only achieves **[0.87](https://www.medrxiv.org/content/10.1101/2022.09.24.22280071v1)** DSC score. None of these advanced benchmarks have achieved 0.894 reported in Totalsegmentator. Therefore, it is sensible to assume the ground truth is biased to the nnU-Net architecture (including iliopsoas that you mentioned and many other classses). This is the reason why we do not make comparisons with Totalsegmentator.
>
> 2. Due to the potential label biases, whenever TotalSegmentator is employed for benchmarking, nnU-Net and models building upon nnU-Net would always outperform other segmentation architectures (e.g., UNETR, TransUNet, SwinUNet, etc.). This observation has also been made in several publications that used the TotalSegmentator dataset. For example, Table 3 in [[Huang et al., 2023](https://arxiv.org/abs/2304.06716)] showed that nnFormer, UNETR, and Swin UNETR were all outperformed by nnU-Net and models building upon nnU-Net in TotalSegmentator. More importantly, the average DSC achieved by our model on TotalSegmentator is also much higher than [[Huang et al., 2023](https://arxiv.org/abs/2304.06716)] as compared in the following table.
>
> | method &nbsp; &nbsp; &nbsp; &nbsp; &nbsp; &nbsp; &nbsp; &nbsp; &nbsp;  | organ &nbsp; &nbsp; &nbsp; | vertebrae &nbsp; &nbsp; &nbsp; | cardiac &nbsp; &nbsp; &nbsp; | muscle &nbsp; &nbsp; &nbsp; |
> |  ----  | ----  |  ----  |  ----  |  ----  |
> | [Huang et al., 2023](https://arxiv.org/abs/2304.06716) | 89.82 | 90.43 | 90.89 | 88.83 |
> | Ours | **92.09** | **91.29** | **92.21** | **95.40** |
>
> Therefore, we believe that the segmentation results in TotalSegmentator reported in our paper are **compelling** and should be considered a **faithful** benchmark.
>
> ---
>
> **Reference**
>
> - Wasserthal, Jakob, Hanns-Christian Breit, Manfred T. Meyer, Maurice Pradella, Daniel Hinck, Alexander W. Sauter, Tobias Heye et al. "Totalsegmentator: Robust segmentation of 104 anatomic structures in ct images." *Radiology: Artificial Intelligence* (2023).
> - Huang, Ziyan, Haoyu Wang, Zhongying Deng, Jin Ye, Yanzhou Su, Hui Sun, Junjun He et al. "STU-Net: Scalable and Transferable Medical Image Segmentation Models Empowered by Large-Scale Supervised Pre-training." *arXiv* (2023).

---

> ### Author Response · Authors · 2023-11-22
> **Response to Reviewer bY7Q (6/6)**
>
> > **Q6.** The paper does not clearly explain how the three expert radiologists collaborated to clean the data, including how they worked together, established uniform standards, and how they corrected tumor masks in the pseudo labels. What was the time cost involved, and so on?
>
> Thank you very much for the suggestion. We have now enclosed this information in the revised Section 3.1 and Appendix B.3.
>
> **Automated organ annotations.** Our annotation pipeline involved an interactive segmentation approach, an integration of AI algorithms and human expertise, which premises to improve the efficiency while upholding high-quality annotations. *One senior radiologist* revised the annotations predicted by our AI models, and in turn, the AI models improved their predictions by learning from these revised annotations. This interactive process continued to enhance the quality of annotations until no major revision is needed. Subsequently, *five junior radiologists* examine the final visualizations for accuracy (examples of the rendered images are illustrated in Appendix B.3 Figure 7). The junior radiologists were responsible for reviewing the correctness of the annotations and marking the patient ID for any major discrepancies. Such cases are then reviewed by the senior radiologist. Our uniform annotation standards, largely overlapping with those in [[Ma et al., FLARE 2022](https://arxiv.org/abs/2308.05862)], require trained radiologists to spend approximately 30–60 minutes annotating each organ in a three-dimensional CT volume.
>
> **Automated (pseudo) tumor annotations.** We have established uniform annotation standards for tumors, with both senior and junior radiologists actively refining and adhering to these guidelines.
>
> - Liver tumors: Liver tumors include primary tumor lesions and metastases in the liver. Annotations should encompass the entire tumor, including any invasive parts, necrosis, hemorrhage, fibrous scars, and calcifications. Healthy areas or unrelated lesions are not included.
>
> - Kidney tumors: Kidney tumors include both benign and malignant tumor lesions growing in the kidneys. The entire tumor and its invasive parts to surrounding areas, plus internal changes like necrosis and calcification, should be annotated. Exclude healthy structures.
>
> - Pancreatic tumors: Pancreatic tumors include all benign and malignant tumor lesions growing in the pancreas. Annotations cover the whole tumor and its invasive growth into adjacent areas, including changes like cysts, necrosis, and calcification. Exclude healthy structures.
>
> - Colon tumors: Colon tumors include all benign and malignant tumor lesions developing from the colon wall. The entire tumor and its invasion into nearby structures, along with internal changes like necrosis, should be annotated, excluding healthy areas.
>
> - Hepatic vessel tumors: Hepatic vessel tumors include all primary tumor lesions developing from the intrahepatic vessel wall and tumor thrombus in intrahepatic vessels. Annotations should include the tumor within the vessels, excluding external parts and unrelated lesions.
>
> Overall, our ImageNetCT-9K dataset offers **51.8K** pseudotumor masks visually inspected by radiologists, though without biopsy confirmation. While these masks lack pathological validation, we anticipate they will serve as a valuable foundation for expanding precise tumor annotations in future research.
>
> ---
>
> > **Q7.** The "novel datasets" claimed in Table 3 is inappropriate. It should be referred to as the external dataset.
>
> Thanks for your suggestion, we have revised the “external dataset” in Table 3 in the new version.
>
> ---
>
> > **Q8.** This paper uses internal medical data for training, so it may need ethical approval for use.
>
> The internal medical data has received IRB approval for use. In Appendix B.1 Table 5, we've detailed the source and permissions for data release. Our approach involves disseminating only the annotations of the CT volumes, which users can combine with the original CT volumes obtained from their original sources. All data created and licensed out by us will be in separate files, ensuring no modifications to the original CT volumes. Legal consultations confirm our permission to distribute these **annotations** under the licenses of each dataset. Upon acceptance of the paper, we will release the entire ImageNetCT-9K dataset to the public. This dataset will provide **296K** organ/tumor masks and **3.7M** annotated images that are taken from **68** hospitals worldwide. And this dataset will continue to expand with the collective effort from the community.
>
> ---
>
> **Reference**
>
> - Ma, Jun, Yao Zhang, Song Gu, Cheng Ge, Shihao Ma, Adamo Young, Cheng Zhu et al. "Unleashing the strengths of unlabeled data in pan-cancer abdominal organ quantification: the flare22 challenge." *arXiv* (2023).

---

### Official Review · Reviewer_esgA · 2023-11-02

**Soundness:** 3 good
**Presentation:** 2 fair
**Contribution:** 3 good
**Rating:** 6
**Confidence:** 4

**Summary:**

This paper evaluates supervised and self-supervised feature learning approaches on 3D CT data. The work first contributes a dataset of CT scans (9000 samples) of different organs, and taken from different hospitals. Next, the authors train different segmentation learning models on this data, both supervised and self-supervised, and report general trends such as sample size efficiency and transferability. The main conclusion is that there is a benefit to having supervised 3D datasets, and that self-supervision can be inferior despite the recent successes of self-supervised models in the vision community. The authors will make the dataset and trained models public.

**Strengths:**

- The dataset that the authors collected seems like it will be a valuable resource for researchers, particularly in medical imaging. The data can serve to train general-purpose 3D features and comes with annotations.

- I do like the point that self-supervision has its limits and that there is a benefit to simply having large supervised data. This is particularly relevant with the current interest in the vision community on self-supervised learning representations.

- Experiments are reasonable and include sufficient prior models.

**Weaknesses:**

- The title may be a little misleading. If I understood it correctly, the word "transfer" in the title is not referring to transfer learning in this case, but simply asking whether supervision is also good for 3D data as it has been for 2D data. When first reading the title, I thought you were exploring transferring 2D supervised models to 3D tasks. Others may also make that incorrect assumption.

- The use of the word "ImageNet" in the dataset name may want to be reconsidered. Besides being a large dataset with a variety of anatomy, the link to ImageNet is a bit weak and may also suggest properties that the dataset does not have (e.g., per-image classification labels).

- CTs are very specific types of 3D data -- it's difficult to make a claim for all 3D data from these experiments alone. I think the paper could be improved by focusing the message more narrowly, perhaps on medical imaging segmentation.

- To me, the results are not surprising -- if you have a such a large dataset with rich segmentation labels, then it should do better than self-supervision. This is in contrast to classification, where this may not be true. I think the main point of the paper should be that this is a new dataset that will be valuable to the community, not that rich supervision is useful in segmentation.

**Questions:**

1. Is there anything surprising from the results? I think it is clear that if you have rich segmentation labels you can learn a better segmentation model than via self-supervised learning.

---

> ### Author Response · Authors · 2023-11-21
> **Response to Reviewer esgA (1/2)**
>
> Thank you for recognizing the value of our dataset, the quality of our presentation, and the extensiveness of our experiments.
>
> ---
>
> > **Q1.** The title may be a little misleading. If I understood it correctly, the word "transfer" in the title is not referring to transfer learning in this case, but simply asking whether supervision is also good for 3D data as it has been for 2D data. When first reading the title, I thought you were exploring transferring 2D supervised models to 3D tasks. Others may also make that incorrect assumption.
>
> The word “transfer” in the title refers to transfer learning, a procedure that involves two stages. **Firstly**, a model is pre-trained on a pretext task, specifically organ segmentation in our paper. **Secondly**, the model is transferred (fine-tuned) to multiple target tasks, including new datasets, classes, and tasks. The target task performance can indicate whether the model pre-training is helpful compared with (1) learning the model from scratch and (2) other publicly available pre-trained models. In this paper, we have assessed the transfer learning ability of models under **five distinct settings**.
> 1. Transfer to different datasets (domains) to segment the same organs—classes that were used for pre-training (Appendix E.2 Table 10).
> 2. Transfer to segmentation tasks of organs, muscles, vertebrae, and cardiac structures—classes that were not used for pre-training (revised Table 4; Appendix E.3 Table 11).
> 3. Transfer to segmentation tasks of pancreatic tumor segmentation—more challenging classes that were not used for pre-training (revised Table 4; Appendix E.3 Table 11).
> 4. Transfer to few-shot segmentation tasks using only a limited number of annotated CT volumes—classes that were not used for pre-training (Figure 1; Figure 2b; Appendix D.1 Figure 9).
> 5. *New:* Transfer to classification tasks that identify fine-grained tumors, including PDAC, Cyst, and PanNet in JHH (Figure 3).
>
> Thank you for your valuable feedback regarding our title. We agree the previous title can be confusing and de-appreciate our contribution. We have now revised it to **“How Well Do Supervised 3D Models Transfer to Medical Imaging Tasks?”** and are actively considering further refinements for greater clarity. One of the contributions of our study, as described in Section 3.2, is the systematic benchmarking of transfer learning performance, particularly from 3D segmentation tasks to a wider range of 3D imaging tasks. Another contribution is the creation of large-scale, per-voxel annotated ImageNetCT-9K, which in turn, makes the transfer learning benchmarking possible.
>
> ---
>
> > **Q2.** The use of the word “ImageNet” in the dataset name may want to be reconsidered. Besides being a large dataset with a variety of anatomy, the link to ImageNet is a bit weak and may also suggest properties that the dataset does not have (e.g., per-image classification labels).
>
> Our objective in developing ImageNetCT-9K is to drive algorithmic advancements and set new benchmarks in the field of 3D medical imaging. In many ways, our dataset echoes the early days of ImageNet, as both datasets emerged at times when large-scale data, diverse classes, and detailed labels were sparse in their respective fields. The limitations of publicly available datasets have been summarized with statistics in Appendix B.1 Table 5 and Figure 5.
>
> Segmentation is often conceptualized as per-voxel classification. In the medical domain, segmentation holds the same fundamental importance as classification does in general computer vision [[Ma & Wang, Nature Methods 2023](https://www.nature.com/articles/s41592-023-01885-0)]. We bet that ImageNet-like datasets in the medical domain should be formed as per-voxel segmentation labels. Our dataset aligns with this vision by providing per-voxel labels, offering a level of detail far surpassing ImageNet's per-image labels. Concretely, the per-voxel labels in our dataset (**272.7B annotated voxels**) are much more extensive than the per-image labels in ImageNet (**14M annotated images**).
>
> We really appreciate your suggestions and are actively looking for better names for our dataset to deliver our vision more precisely for the medical imaging field and to inspire further research endeavors towards this end.
>
> ---
>
> **Reference**
>
> - Ma, Jun, and Bo Wang. "Towards foundation models of biological image segmentation." *Nature Methods* (2023).

---

> ### Author Response · Authors · 2023-11-21
> **Response to Reviewer esgA (2/2)**
>
> > **Q3.** CTs are very specific types of 3D data -- it's difficult to make a claim for all 3D data from these experiments alone. I think the paper could be improved by focusing the message more narrowly, perhaps on medical imaging segmentation.
>
> We completely agree with your comments and have now revised our title, abstract, introduction, and conclusion to narrow down our current scope to 3D medical imaging.
>
> Given that our dataset includes detailed per-voxel annotations for 25 organs and tumors, it enables the automatic generation of 3D shape representations. These representations can be formatted as point clouds, voxel occupancy grids, meshes, and implicit surface models (e.g., signed distance functions), each catering to different algorithmic needs. We anticipate our dataset could be useful for a variety of other 3D medical vision tasks [[Li et al., 2023](https://arxiv.org/abs/2308.16139)], such as pose estimation, surface reconstruction, depth estimation, etc. Since these studies go far beyond the scope of the current manuscript and our expertise, we would like to leave the investigation as an independent work in the future. The potential applications and implications of our dataset have now been included in the future work section and Appendix F.4.
>
> ---
>
> > **Q4.** Is there anything surprising from the results? I think it is clear that if you have rich segmentation labels you can learn a better segmentation model than via self-supervised learning. I think the main point of the paper should be that this is a new dataset that will be valuable to the community, not that rich supervision is useful in segmentation.
>
> Thank you for recognizing the value of our dataset. As detailed in **Q1**, we have systematically benchmarked the transfer learning ability of public models and are preparing to release a suite of models that are pre-trained on our extensive, annotated dataset. The download links are available in our common responses. These two contributions align closely with the conference of *Learning Representation* (IC*LR*) and our submission is categorized into the *“datasets and benchmarks”* primary area.
>
> The debate between the effectiveness of representation learning using human-annotated data (supervised pre-training) versus raw data (self-supervised pre-training) has been a longstanding topic in general computer vision, as we review in Section 2. As acknowledged by Reviewer fMfJ, *this paper concluded on the debate of whether self-supervised or supervised pre-training leads to better performance and data efficiency. This debate had not be resolved without the invention of a fully-annotation dataset of such scale.* Our constructed dataset enables this crucial comparison and promises to be a valuable asset for future algorithm benchmarking in the field.
>
> Based on our comparison, two observations are particularly surprising. Please note that in this comparison, the model is transferred to the classes and datasets that differ from those used for pre-training.
>
> 1. Efficiency in pretext task: Supervised pre-training requires **99.6%** fewer data and **96.5%** less computation than self-supervised pre-training. The model trained with 21 CT volumes, 672 masks, and 40 GPU hours shows a transfer learning ability similar to that trained with 5,050 CT volumes and 1,152 GPU hours. See details in Figure 2a.
>
> 2. Efficiency in target task: Supervised pre-training requires **50%** fewer manual annotations for fine-grained tumor classification than self-supervised pre-training. This is particularly critical for tumor imaging tasks because annotating tumors requires much more effort and often relies on the availability of pathology reports. See details in Figure 2b and Appendix D.1 Figure 9.
>
> ---
>
> **Reference**
>
> - Li, Jianning, Antonio Pepe, Christina Gsaxner, Gijs Luijten, Yuan Jin, Narmada Ambigapathy, Enrico Nasca et al. "MedShapeNet--A Large-Scale Dataset of 3D Medical Shapes for Computer Vision." *arXiv* (2023).

---

### Official Review · Reviewer_fMfJ · 2023-11-03

**Soundness:** 3 good
**Presentation:** 3 good
**Contribution:** 3 good
**Rating:** 8
**Confidence:** 3

**Summary:**

This paper presents a new large-scale computed tomography (CT) dataset for medical image segmentation, which is so-far the one with the highest number of annotated scans. The authors, employing this benchmark, draw several insights where most of them are revealed for the first time. More specifically, the authors show that supervised pre-training is more effective and efficient compared with self-supervised counterpart given similar training circumstances. Their released models can effectively serve as a foundation model for transfer learning, helping to reduce the computational load and improve the segmentation accuracy.

**Strengths:**

I really enjoy reading this paper and the contribution it brings. First, the authors would release a large-scale volumetric segmentation dataset with unprecedented number of pixelwisely labeled ground truth. This dataset comes from some publically available dataset and self-constructed ones with semi-annotated tools and interactive segmentation with radiologists. Second, the paper concluded on the debate of whether self-supervised or supervised pre-training lead to better performance and data efficiency. This debate had not be resolved without the invention of a fully-annotation dataset of such scale. Third, the authors release their pre-trained models so that one can easily fine-tune the model efficiently. This can also have a good impact for the whole community.

This paper is well structure and written. The notation is clear, and the experimental setups are carefully noted.

Experimental results are intensive and convincing. I checked the attached code and it seems to be solid.

**Weaknesses:**

I do not remark any major issue as the drawback of this paper.

**Questions:**

I haven't had many questions regarding this paper.

- Could the authors explain why the performance "scratch" out-performed most of the pre-training method in Tab. 2?

---

> ### Author Response · Authors · 2023-11-21
> **Response to Reviewer fMfJ**
>
> We are grateful for your accolades on our contributions, *“...a foundation model for transfer learning…”*, *“...concluded on the debate…”*, *“...the invention of a fully-annotation dataset of such scale”*, and potential impact, *“one can easily fine-tune the model efficiently”*.
>
> ---
>
> > **Q1.** Could the authors explain why the performance "scratch" out-performed most of the pre-training method in Tab. 2?
>
> The goal of Table 2 is to provide a practical benchmark for the transfer learning ability of readily available pre-trained models. Our intent is not to compare the specific pre-training methodologies of each model for two primary reasons. **Firstly**, the majority of researchers tend to fine-tune pre-existing models rather than retrain them from scratch due to convenience and accessibility. **Secondly**, reproducing these models would require specialized hyper-parameter tuning and varied computational resources. For example, models like Swin UNETR [[Tang et al., CVPR 2023](https://openaccess.thecvf.com/content/CVPR2022/papers/Tang_Self-Supervised_Pre-Training_of_Swin_Transformers_for_3D_Medical_Image_Analysis_CVPR_2022_paper.pdf)] are pre-trained using large-scale GPU clusters at NVIDIA, making them challenging for us to faithfully retrain. Considering both practical user scenarios and computational constraints, we decided to directly use their released models and fine-tune them with consistent settings on the same datasets.
>
> Using existing pre-trained models can inevitably lead to certain problems. For example, the U-Net family has seen numerous variations over the years [[Siddique et al., 2021](https://ieeexplore.ieee.org/abstract/document/9446143)]. Pre-trained models released before 2021 typically employed a basic version of U-Net (e.g., [[Zhou et al., 2019](https://www.ncbi.nlm.nih.gov/pmc/articles/PMC7405596/)] and [[Chen et al., 2019](https://arxiv.org/abs/1904.00625)] in Table 2). On the other hand, our U-Net benefits from a more advanced code base, thanks to the [MONAI](https://monai.io/) platform at NVIDIA, which includes enhanced architectures and advanced training optimization strategies. Consequently, our U-Net, even trained from scratch, is capable of surpassing the performance of these older baseline models.
>
> ---
>
> **Reference**
>
> - Tang, Yucheng, Dong Yang, Wenqi Li, Holger R. Roth, Bennett Landman, Daguang Xu, Vishwesh Nath, and Ali Hatamizadeh. "Self-supervised pre-training of swin transformers for 3d medical image analysis." *CVPR* (2022).
> - Siddique, Nahian, Sidike Paheding, Colin P. Elkin, and Vijay Devabhaktuni. "U-net and its variants for medical image segmentation: A review of theory and applications." *Ieee Access* (2021).
> - Zhou, Zongwei, Vatsal Sodha, Md Mahfuzur Rahman Siddiquee, Ruibin Feng, Nima Tajbakhsh, Michael B. Gotway, and Jianming Liang. "Models genesis: Generic autodidactic models for 3d medical image analysis." *MICCAI* (2019).
> - Chen, Sihong, Kai Ma, and Yefeng Zheng. "Med3d: Transfer learning for 3d medical image analysis." *arXiv* (2019).

---

### Author Response · Authors · 2023-11-21
**General Response: Our Clarifications, Common Questions, and Major Updates (2/2)**

**II. Common questions**

- **Releasing code and models?**
We have already attached our source code to the supplementary material (acknowledged by Reviewer fMfJ). With the growing trend of using pre-trained models, there is a need for standardized, accessible approaches to sharing public model weights. In line with this, we have released a suite of pre-trained models summarized in the table below for reviewers and the public audience. Releasing pre-trained foundation models should be considered a marked contribution as they offer an alternative way of knowledge sharing while protecting patient privacy [[Zhang et al. MEDIA](https://www.sciencedirect.com/science/article/pii/S1361841523002566)].

| model backbone &nbsp; &nbsp; &nbsp; &nbsp; &nbsp;| parameters &nbsp; &nbsp; | download &nbsp; &nbsp; |
|:--------| :-------------|:-------------|
| U-Net |19.08M| [link](https://drive.google.com/file/d/1-NoUA3dJhZUncFdSLiQdh4a6iIYaOmU7/view?usp=drive_link) |
| Swin UNETR |62.19M| [link](https://drive.google.com/file/d/1aByA0F4FWS5EWrSl3oazzlixD-v-WqTB/view?usp=drive_link) |
|SegResNet|470.13M|[link](https://drive.google.com/file/d/1Du8AOLUklJfE5XHfBY_jZivcfPIyI5Pt/view?usp=drive_link)|

- **Ethics concerns of data and annotations?**
In Appendix B.1 Table 5, we've detailed the source and permissions for data release. Our approach involves disseminating only the annotations of the CT volumes, which users can combine with the original CT volumes obtained from their original sources (Reviewers bY7Q and osvs). All data created and licensed out by us will be in separate files, ensuring no modifications to the original CT volumes. Legal consultations confirm our permission to distribute these annotations under the licenses of each dataset. Upon acceptance of the paper, we will also release the entire ImageNetCT-9K dataset to the public. This dataset will provide **296K** organ/tumor masks and **3.7M** annotated images that are taken from **68** hospitals worldwide. And this dataset will continue to expand with the collective effort from the community.

- **Can models, pre-trained on 3D medical data, transfer to other 3D vision tasks?**
In our revised manuscript, we have updated both the abstract and introduction to more accurately reflect our research findings in 3D medical image analysis (Reviewers esgA and bY7Q). The potential of transferring our pre-trained models to alternative imaging modalities and broader 3D vision tasks has been discussed in the future work section.

---

**III. Major updates**

1. We have stressed the significance of our pre-trained models in transfer learning using the five experimental settings clarified above (see revised Section 4; Appendix A).

2. We have detailed the procedure for constructing and annotating our dataset, combining the best of three expert radiologists and an AI algorithm using a uniform standard. (see revised Section 3.1; Appendix B.3).

3. We have justified the performance obtained by "scratch" and other publicly available pre-trained models to avoid potential confusion (see revised Table 2; Appendix C.2 Table 8).

4. We have clarified the experimental settings between self-supervised and supervised pre-training adopted in our comparison (see revised Section 3.2 and Section 4).

5. We have justified with explicit evidence that our pre-trained models could address the domain transfer problem. More discussions about the direct inference results are presented in the Table 3 caption.

6. We have included more related works, elaborated our future work, and revised figures, tables, and main text to provide a more coherent, comprehensive, and compact presentation (see revised Section 2; Section 5; Appendix F.4).

In addition, we have also taken this revision opportunity to further improve our manuscript in the following aspects.

7. We have investigated the **cross-task** ability of our pre-trained models by transferring from organ segmentation to fine-grained tumor classification. The distance between the two tasks is much larger than transferring among segmentation tasks, so we hope the new result in Figure 3 can further strengthen the practical impact of our pre-trained models.

8. We have shown the **annotation efficiency** achieved by our pre-trained models. This is particularly critical for tumor imaging tasks because annotating tumors requires much more effort and often relies on the availability of pathology reports. Our results in Figure 2b and Appendix D.1 Figure 9 suggest that fine-tuning our models can reduce annotation costs for organ segmentation and fine-grained tumor classification by 50%.

---

**Reference**

- Zhang, Shaoting, and Dimitris Metaxas. "On the Challenges and Perspectives of Foundation Models for Medical Image Analysis." *Medical image analysis* (2023).

---

### Author Response · Authors · 2023-11-21
**General Response: Our Clarifications, Common Questions, and Major Updates (1/2)**

We extend our thanks to all reviewers for recognizing our **two practical contributions**: the construction of a large-scale, per-voxel annotated dataset and the benchmark of a suite of segmentation foundation models, leading to **one technical contribution**: spark the debate of whether self-supervised or supervised pre-training lead to better performance and data efficiency.

In this response, we will clarify some of our experimental settings, address general questions from reviewers, and outline the major updates in the manuscript. Apart from common responses, we will also point-by-point address specific questions raised by each reviewer. The revised manuscript will be attached soon, with all new content marked in cyan for ease of track. *The index for Sections, Figures, and Tables are based on the revised manuscript*.

---

**I. Clarification**

We would like to clarify our transfer learning settings in this paper. We assess the transfer learning ability of models—supervised pre-trained by organ segmentation—under **five distinct settings**.
1. Transfer to external datasets (domains) to segment the same organs—classes that were used for pre-training (Appendix E.2 Table 10).
2. Transfer to segmentation tasks of organs, muscles, vertebrae, and cardiac structures—classes that were not used for pre-training (revised Table 4; Appendix E.3 Table 11).
3. Transfer to segmentation tasks of pancreatic tumor segmentation—more challenging classes that were not used for pre-training (revised Table 4; Appendix E.3 Table 11).
4. Transfer to few-shot segmentation tasks using only a limited number of annotated CT volumes—classes that were not used for pre-training (Figure 1; Figure 2b; Appendix D.1 Figure 9).
5. *New:* Transfer to classification tasks that identify fine-grained tumors, including PDAC, Cyst, and PanNet in JHH (Figure 3).

This evaluation protocol has been widely adopted to assess transfer learning ability in medical imaging [[Zhou et al., MEDIA](https://www.sciencedirect.com/science/article/pii/S1361841520302048); [Tang et al., CVPR 2023](https://openaccess.thecvf.com/content/CVPR2022/papers/Tang_Self-Supervised_Pre-Training_of_Swin_Transformers_for_3D_Medical_Image_Analysis_CVPR_2022_paper.pdf); [Jiang et al., CVPR 2023](https://openaccess.thecvf.com/content/ICCV2023/papers/Jiang_Anatomical_Invariance_Modeling_and_Semantic_Alignment_for_Self-supervised_Learning_in_ICCV_2023_paper.pdf)] and computer vision [[He et al., CVPR 2022](https://openaccess.thecvf.com/content/CVPR2022/papers/He_Masked_Autoencoders_Are_Scalable_Vision_Learners_CVPR_2022_paper.pdf); [Zhai et al., CVPR 2022](https://openaccess.thecvf.com/content/CVPR2022/papers/Zhai_Scaling_Vision_Transformers_CVPR_2022_paper.pdf)]. In addition, we plan to assess the transfer learning ability across imaging modalities (e.g., MRI; suggested by Reviewer mSzX) and broader 3D vision tasks (suggested by Reviewers esgA and bY7Q).

---

**Reference**

- Zhou, Zongwei, Vatsal Sodha, Jiaxuan Pang, Michael B. Gotway, and Jianming Liang. "Models genesis." *Medical image analysis* (2021).

- Tang, Yucheng, Dong Yang, Wenqi Li, Holger R. Roth, Bennett Landman, Daguang Xu, Vishwesh Nath, and Ali Hatamizadeh. "Self-supervised pre-training of swin transformers for 3d medical image analysis." *CVPR* (2022).

- Jiang, Yankai, Mingze Sun, Heng Guo, Xiaoyu Bai, Ke Yan, Le Lu, and Minfeng Xu. "Anatomical Invariance Modeling and Semantic Alignment for Self-supervised Learning in 3D Medical Image Analysis." *ICCV* (2023).

- He, Kaiming, Xinlei Chen, Saining Xie, Yanghao Li, Piotr Dollár, and Ross Girshick. "Masked autoencoders are scalable vision learners." *CVPR* (2022).

- Zhai, Xiaohua, Alexander Kolesnikov, Neil Houlsby, and Lucas Beyer. "Scaling vision transformers." *CVPR* (2022).

---

### Author Response · Authors · 2023-11-23
**Any Last-minute Feedback and Big Thanks to All the Reviewers**

Dear all reviewers,

As the discussion period nears its conclusion, we wanted to reach out for any last-minute feedback you might have. We are very grateful for your insightful suggestions and have made every effort to address your concerns in our revised manuscript. The revised manuscript is attached, with all new content marked in cyan for ease of track.

We have endeavored to prepare a significantly better ICLR paper, taking all the critiques into careful consideration, including new references, figures, and text; and rewriting existing text to provide a more accurate, comprehensible, and compact presentation within the page limit.

Wishing you a Happy Thanksgiving!

Authors

---

### Public Comment · ~Nina_Montaña-Brown1 · 2024-04-30

Great work on the dataset. Was there any reason not to include the SARAMIS (https://openreview.net/forum?id=SEU9m9NReo&noteId=SEU9m9NReo) dataset for the verification of transfer learning capabilities (Tab 3), which includes radiologist verified segmentation labels over 106 classes for the AMOS and AbdomenCT-1k datasets?

Congratulations on the oral presentation!

---

> ### Public Comment · ~Zongwei_Zhou1 · 2024-05-02
>
> Dear Nina,
>
> Thank you for introducing us to the valuable SARAMIS dataset. We were previously unaware of its existence until your helpful introduction. We have successfully downloaded SARAMIS and are excited to explore its potential in our future publications.
>
> Best regards,
>
> Zongwei

---

### Meta-Review · Area_Chair_Qor7 · 2023-12-02

**Metareview:**

This submission receives a set of strong scores: 8, 6, 6, 8, 6. All 5 reviewers agree to accept the paper.

This submission presents a significant contribution to the field of machine learning for 3D medical imaging. The authors introduce ImageNetCT-9K, a large dataset comprising ~9K CT volumes with high-quality voxel-level annotations. They demonstrate the utility of this dataset through extensive experiments, showing that the model trained with only 21 CT volumes (672 masks & 40 GPU hours) has a transfer learning ability similar to the model trained with 5,050 CT volumes with 1,152 GPU hours. Also, the authors pointed out that the transfer learning ability of supervised models can further scale up with larger annotated datasets, achieving better performance than all existing pre-trained models, irrespective of their pre-training methodologies or data sources. The paper's strengths lie in its valuable dataset, thorough experiments, and the great potential for advancing supervised/transfer learning methods in 3D medical imaging.

The reviewer bY7Q noted that ethical review may be warranted for this submission to ensure adequate protections for privacy, security, and safety. Given that the work utilizes medical data for training and proposes open-sourcing the dataset contingent upon acceptance, further ethical vetting could be prudent. As only the authors currently have access to the model and data, they should diligently verify adherence to pertinent ethical codes and requirements.

**Justification For Why Not Higher Score:**

- Need More Comparative Analysis with Other Large Datasets: a reviewer pointed out the absence of a detailed comparison with similar large-scale datasets like UniverSeg, which could provide a more comprehensive understanding of the dataset's uniqueness and its contribution to the field.

- Limited Exploration of Pre-training Strategies: a reviewer highlighted the paper's limited exploration into different pre-training strategies, particularly the potential of category-guided InfoNCE loss, which might offer insights into alternative approaches for model training.

- Generalization to Other Modalities: While SUPREM shows promising results, its generalization to modalities beyond CT, such as MRI, wasn't thoroughly explored, as noted by a reviewer. This limitation restricts the broader applicability of the findings.

**Justification For Why Not Lower Score:**

- Significant Contribution of Dataset: The IMAGENETCT-9K dataset is unprecedented in scale, providing a substantial leap in annotations and diversity, addressing a crucial gap in 3D medical imaging as recognized by all reviewers.

- Efficiency of Supervised Pre-training: The study convincingly demonstrates the efficiency of supervised pre-training over self-supervised methods, as shown by the model's performance with significantly less training data and computational resources.

- Broad Impact and Open Science: The decision to make the dataset and models publicly available aligns with open science principles, likely to foster further research and advancements in the field, a point well-received by the reviewers.

- Positive Reception by Reviewers: Despite some criticisms, all reviewers acknowledged the strengths of the paper, with none suggesting a rejection. The paper's contribution in sparking a debate on supervised versus self-supervised pre-training in medical imaging was particularly appreciated.

---

### Decision · Program_Chairs · 2024-01-16

Accept (oral)